# Data-independent acquisition method for ubiquitinome analysis reveals regulation of circadian biology

Fynn M. Hansen[1], Maria C. Tanzer[1], Franziska Brüning [1,2], Isabell Bludau[1], Che Stafford[3], Brenda A. Schulman [4], Maria S. Robles [2✉], Ozge Karayel [1✉] & Matthias Mann [1✉]

Protein ubiquitination is involved in virtually all cellular processes. Enrichment strategies employing antibodies targeting ubiquitin-derived diGly remnants combined with mass spectrometry (MS) have enabled investigations of ubiquitin signaling at a large scale. However, so far the power of data independent acquisition (DIA) with regards to sensitivity in single run analysis and data completeness have not yet been explored. Here, we develop a sensitive workflow combining diGly antibody-based enrichment and optimized Orbitrap-based DIA with comprehensive spectral libraries together containing more than 90,000 diGly peptides. This approach identifies 35,000 diGly peptides in single measurements of proteasome inhibitor-treated cells – double the number and quantitative accuracy of data dependent acquisition. Applied to TNF signaling, the workflow comprehensively captures known sites while adding many novel ones. An in-depth, systems-wide investigation of ubiquitination across the circadian cycle uncovers hundreds of cycling ubiquitination sites and dozens of cycling ubiquitin clusters within individual membrane protein receptors and transporters, highlighting new connections between metabolism and circadian regulation.

[1] Department of Proteomics and Signal Transduction, Max Planck Institute of Biochemistry, Martinsried, Germany. [2] Institute of Medical Psychology, Faculty of Medicine, LMU, Munich, Germany. [3] Gene Center and Department of Biochemistry, Ludwig-Maximilians-Universität München, Munich, Germany. [4] Department of Molecular Machines and Signaling, Max Planck Institute of Biochemistry, Martinsried, Germany. ✉email: charo.robles@med.uni-muenchen. de; karayel@biochem.mpg.de; mmann@biochem.mpg.de

Ubiquitination is a reversible and highly versatile post-translational modification (PTM) involved in virtually all cellular processes. A ubiquitin conjugation cascade, involving ubiquitin activating (E1), conjugating (E2), and ligating (E3) enzymes, mediates the covalent attachment of the 76 amino acid long ubiquitin molecule to a ε-amine group of a lysine residue on a substrate protein. Its removal is mediated by an enzyme family called deubiquitinating enzymes (DUB). Ubiquitin itself can be ubiquitinated N-terminally or via one of its seven lysine residues, giving rise to a plethora of chain topologies, which encode a diverse and specific set of biological functions[1,2]. Deregulation of this highly complex process has been linked to numerous diseases including neurodegenerative diseases[3,4], autoimmunity[5,6], and inflammatory disorders[7–9].

Protein ubiquitination is one of the most widely studied PTMs in the field of mass spectrometry (MS)-based proteomics. However, due to low stoichiometry of ubiquitination and varying ubiquitin-chain topologies, comprehensive profiling of endogenous ubiquitination is challenging and requires one or more enrichment steps prior to MS analysis[10]. Early reports to catalog ubiquitin conjugated proteins from yeast and human described various enrichment methods including the use of epitope-tagged ubiquitin or ubiquitin-associated domains (UBA)[11–13]. After trypsinization previously ubiquitinated peptides bear a signature diGly remnant that can be targeted by a specific antibody[14]. Enrichment strategies employing such antibodies have enabled identification of thousands of ubiquitination sites by MS[15–17]. A recently described antibody targets a longer remnant generated by LysC digestion to exclude ubiquitin-like modifications such as NEDD8 or ISG15[18], however, the contribution of diGly sites derived from ubiquitin-like modifications is very low (<6%)[15].

The commercialization of such antibodies has accelerated MS-based ubiquitinome analysis and enabled a variety of quantitative, systems-wide studies[19–23]. However, large-scale analysis of ubiquitination events to study key signaling components remains challenging since in-depth diGly proteome coverage requires relatively large sample amounts and extensive peptide fractionation. These requirements, which largely stem from the low stoichiometry of the modification, come at the expense of throughput, robustness, and quantitative accuracy.

Thus far, ubiquitinome studies have employed data-dependent acquisition (DDA) methods combined with label-free or isotope-based quantification[24]. Recently, data-independent acquisition (DIA) has become a compelling alternative to DDA for proteomics analysis enabling greater data completeness across samples[25–28]. In contrast to intensity-based precursor picking of DDA, DIA fragments all co-eluting peptide ions within pre-defined mass-to-charge ($m/z$) windows and acquires them simultaneously[29]. This leads to more precise and accurate quantification with fewer missing values across samples and higher identification rates over a larger dynamic range. DIA usually requires a comprehensive spectral library, from which the peptides are matched into single-run MS analyses. Recently, superior performance of DIA for sensitive and reproducible MS measurements has also been demonstrated for global protein phosphorylation analysis[30]. Given the central importance of ubiquitination, we here set out to investigate the power of DIA for improving data completeness and sensitivity in a single-run analysis format.

For sensitive and reproducible analysis of the ubiquitin-modified proteome, we here devise a workflow combining diGly antibody-based enrichment with a DIA method tailored to the unique properties of the library peptides and to the linear quadrupole Orbitrap mass analyzer. We acquire extensive spectral libraries that altogether contained more than 90,000 diGly peptides allowing us to reproducibly analyze 35,000 distinct diGly

peptides in a single measurement of proteasome inhibitor-treated cells. The DIA-based diGly workflow markedly improves the number of identifications and quantitative accuracy compared to DDA. To investigate if our method would have advantages in the exploration of biological signaling systems, we first apply it to the well-studied TNF-signaling pathway, where it retrieves known ubiquitination events and uncoveres novel ones. We then extend it to the analysis of circadian post-translational dynamics, so far poorly studied globally with regards to ubiquitination. This uncovers a remarkable extent and diversity of cycling ubiquitination events. These include closely spaced clusters with the same circadian phase, which are likely pointing to novel mechanisms. Together, our design and results establish a sensitive and accurate DIA-based workflow suitable for investigations of ubiquitin signaling at a systems-wide scale.

## Results

**DIA quantification enables in-depth diGly proteome coverage in single-shot experiments**. To obtain a comprehensive, in-depth spectral library for efficient extraction of diGly peptides in single-shot DIA analysis, we treated two human cell lines (HEK293 and U2OS) with a common proteasome inhibitor (10 μM MG132, 4 h). After extraction and digestion of proteins, we separated peptides by basic reversed-phase (bRP) chromatography into 96 fractions, which were concatenated into 8 fractions ("Methods", Supplementary Fig. 1a). Here, we isolated fractions containing the highly abundant K48-linked ubiquitin-chain derived diGly peptide (K48-peptide) and processed them separately to reduce excess amounts of K48-peptides in individual pools, which compete for antibody binding sites during enrichment and interfere with the detection of co-eluting peptides (Supplementary Fig. 1b). We found this to be a particular issue for MG132 treatment, as blockage of the proteasome activity further increases K48-peptide abundance in these samples. The resulting nine pooled fractions were enriched for diGly peptides, which were separately analyzed using a DDA method (PTMScan Ubiquitin Remnant Motif (K-ε-GG) Kit, CST) (Fig. 1a and Supplementary Fig. 1a-b). This identified more than 67,000 and 53,000 diGly peptides in MG132 treated HEK293 and U2OS cell lines, respectively (Fig. 1b). Furthermore, to fully cover diGly peptides of an unperturbed system, we also generated a third library using the same workflow but with untreated U2OS cells (used later for biological applications). This added a further 6000 distinct diGly peptides (Fig. 1b). In total, we obtained 89,650 diGly sites corresponding to 93,684 unique diGly peptides, 43,338 of which were detected in at least two libraries (Fig. 1c, see also source data at PRIDE: PXD019854). To our knowledge, this represents the deepest diGly proteome to date. According to the PhosphositePlus database[31], 57% of the identified diGly sites were not reported before and 7.3% of them had previously been found to be acetylated or methylated, indicating that different PTMs can act on the same sites. Thus, the growing body of diGly sites can help to identify sites of potential PTM crosstalk, an important level of functional regulation of proteins[32].

In possession of these large diGly spectral libraries, we evaluated DIA method settings for best performance in single-shot diGly experiments (Supplementary Data 1). Impeded C-terminal cleavage of modified lysine residues frequently generates longer peptides with higher charge states, resulting in diGly precursors with unique characteristics. Guided by the empirical precursor distributions, we first optimized DIA window widths—the transmission windows that together cover the desired precursor peptide range. This increased the number of identified diGly peptides by 6% (Supplementary Fig. 2a-b). Next, we tested different window numbers and fragment scan resolution settings,

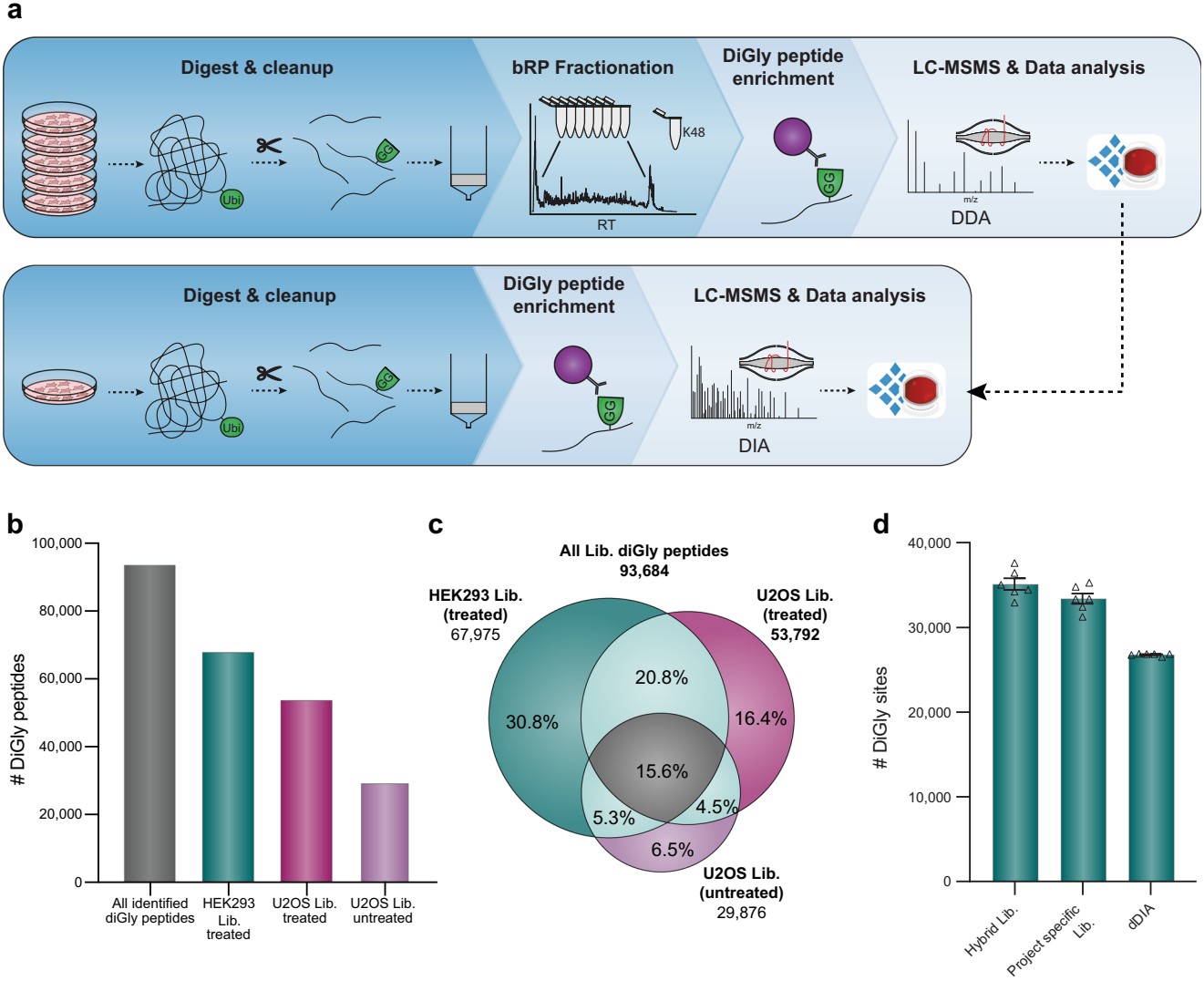

**Fig. 1 In-depth diGly proteomics for DIA identification. a** Experimental workflow for in-depth diGly peptide library construction (upper panel) and our single-run data-independent acquisition (DIA)-based workflow (lower panel). Protein digestion and peptide extraction are followed by basic reversed-phase (bRP) fractionation and diGly peptide enrichment. For library construction, samples were measured by data-dependent acquisition (DDA) and computationally processed (Spectronaut Pulsar). Individual samples are measured by our DIA workflow, including matching against a library for identification (Spectronaut software). **b** Number of identified diGly peptides in three different spectral libraries (MG132 treated HEK293 library—green, MG132 treated U2OS library—violet, U2OS library—light violet, all diGly peptides—gray). **c** Commonly and exclusively identified diGly peptides for different libraries (MG132 treated HEK293 library—green, MG132 treated U2OS library—violet, U2OS library—light violet). **d** Identified diGly sites (mean ± SEM) of MG132 treated HEK293 cells using different DIA library search strategies ($n = 6$, three workflow replicates measured in analytical duplicates). Source data are provided as a Source data file.

to strike an optimal balance between data quality and a cycle time that sufficiently samples eluting chromatographic peaks. We found that a method with relatively high MS2 resolution of 30,000 and 46 precursor isolation windows performed best (13% improvement compared to the standard full proteome method that we started with) (Supplementary Fig. 2c). Furthermore, we determined the optimal antibody and peptide input combination to maximize peptide yield and depth of coverage in single DIA experiments. To mimic endogenous cellular levels, we used peptide input from cells not treated with MG132. From titration experiments, enrichment from 1 mg of peptide material using 1/8th of an anti-diGly antibody vial (31.25 µg) turned out to be optimal ("Methods" and Supplementary Fig. 2d, e). With the improved sensitivity by DIA, only 25% of the total enriched material needed to be injected (Supplementary Fig. 2f).

Using our optimized DIA-based workflow, we identified a remarkable 33,409 ± 605 distinct diGly sites in single measurements of MG132 treated HEK293 samples. This implies that about half of the sites in the deep, cell line-specific spectral library was matched into the single runs. Interestingly, even without using any library, a search of six single runs identified 26,780 ± 59 diGly sites (direct DIA, "Methods"). Finally, employing a hybrid spectral library—generated by merging the DDA library with a direct DIA search—resulted in 35,111 ± 682 diGly sites in the same samples (Fig. 1d, Supplementary Data 2). Compared to recent reports in the literature[24], these numbers double diGly peptide identifications in a single-run format.

**DIA improves diGly proteome quantification accuracy.** To evaluate the reproducibility of the entire DIA-based diGly

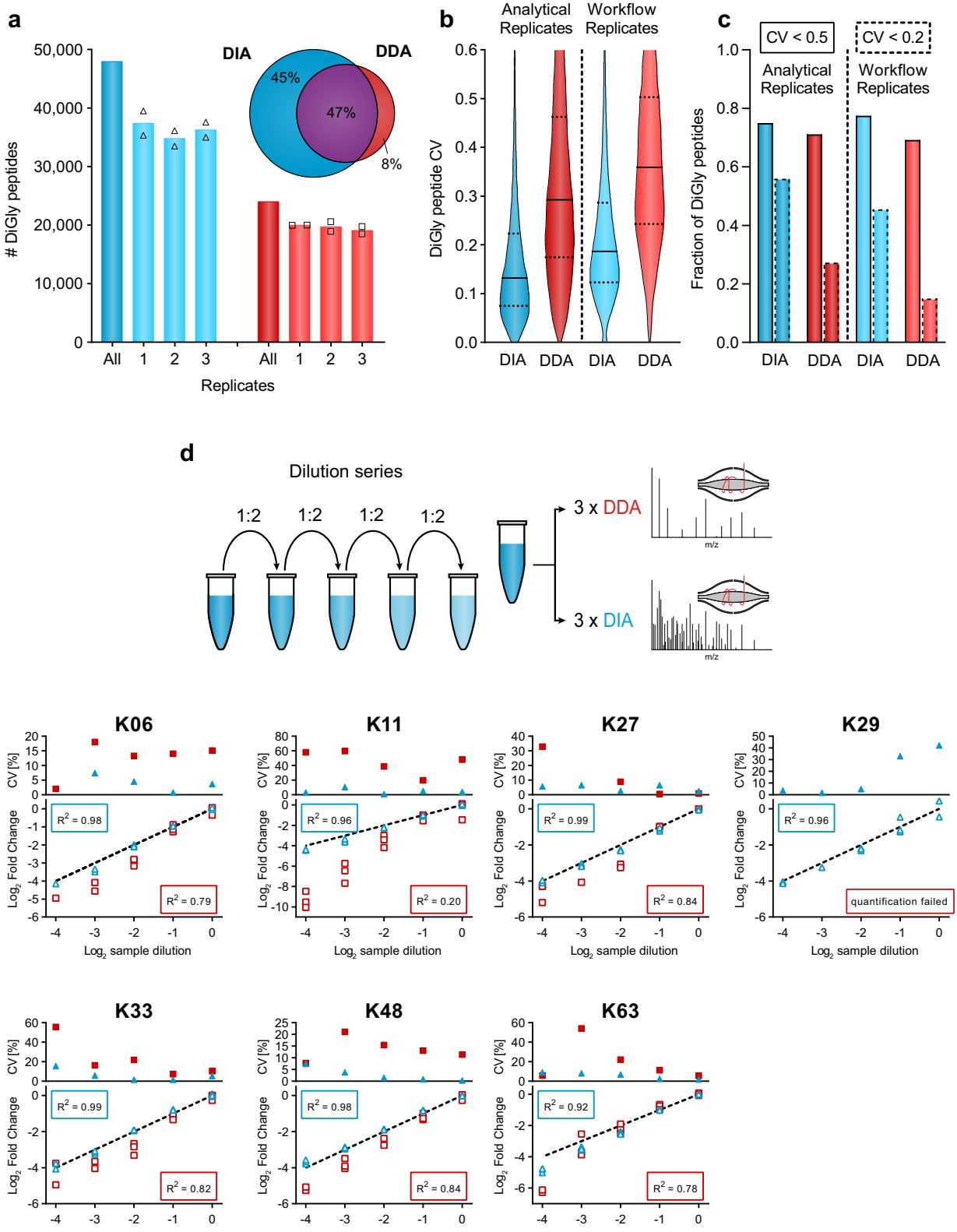

workflow, we used MG132 treated HEK293 cells and performed three independent diGly peptide enrichments followed by DIA analysis in duplicates. This identified around 36,000 distinct diGly peptides in all replicates, 45% and 77% of which had coefficients of variations (CVs) below 20% and 50%, respectively (Fig. 2a–c, Supplementary Data 3). In contrast, a DDA method identified substantially fewer distinct diGly peptides and a smaller percentage with good CVs (20,000 diGly peptides; 15% with CVs <20%; Fig. 2a–c). Overall, the six DIA experiments yielded almost 48,000 distinct diGly peptides, while the corresponding DDA experiments resulted in 24,000 diGly peptides. Furthermore, the improved reproducibility is apparent from the diGly site data matrix, which has considerably fewer missing values (Supplementary Fig. 2g).

**Fig. 2 Accurate and reproducible diGly proteomics for DIA quantification. a** Number of identified diGly peptides (mean, $n = 2$) for data-independent acquisition (DIA, blue, HEK293 hybrid library) and data-dependent acquisition (DDA, red) strategies ($n = 6$, three workflow replicates measured in analytical duplicates). Venn diagram depicts the proportion of shared and exclusively identified diGly sites between DIA and DDA approaches. **b** Coefficient of variation (CV) value distribution for DIA and DDA approaches. Solid and dotted lines denote median and 1st or 3rd quantile, respectively. **c** Fractions of CV values below 50% and 20% are shown with solid and dotted lines, respectively. **d** Dilution series of diGly enriched sample. Plots show individual ubiquitin-chain linkage type peptides measured via DIA (blue) or DDA (red) ($n = 3$). Top panels depict CV values of replicate measurements. Bottom panels show individual measurements compared to the expected dilution depicted as dotted line. $R^2$ values describe the goodness-of-fit of measured values to the expected dilution series (dotted line). Source data are provided as a Source data file.

To further investigate the quantitative precision and accuracy of our method, we turned to ubiquitin-chain linkage derived diGly peptides. These are the most abundant diGly peptides, all ranking in the top 20 by abundance and spanning three orders of magnitude in MS signal (Supplementary Fig. 2h). Diverse chain linkages confer various functions to proteins; hence, accurate quantification is important to decode the cellular roles of different ubiquitin linkage types. We performed a dilution series of a diGly sample and analyzed each dilution sample using both DIA and DDA methods in triplicates. Linear regression of measured vs. expected dilution factors, as a means to directly compare the performance of DIA against DDA, resulted in $R^2$ values higher than 0.92 for all seven chain peptides assessed, much higher than the corresponding values for DDA ($R^2$ 0.20–0.84; Fig. 2d, Supplementary Data 3). Importantly for quantification purposes, the experimentally observed slope for DIA was much closer to 1 than for DDA.

Together, these analytical results establish that the DIA-based workflow substantially increased the number of diGly peptides identified while markedly improving the precision and accuracy of quantification compared to a DDA-based workflow.

**In-depth ubiquitinome analysis of the TNF-signaling pathway.** The pro-inflammatory properties of TNF are heavily regulated by dynamic ubiquitination of its receptor-signaling complex (RSC)[33,34] and global ubiquitinome changes upon TNF stimulation were described previously in a proteomics study[35]. Encouraged by the technical capabilities of our DIA-based diGly workflow, we here aimed to test our DIA-based diGly workflow on this well-studied system, to demonstrate benefits of DIA over DDA based on accurate ubiquitination site quantification and, if possible, to extend the current knowledge of the TNF-regulated ubiquitinome (Fig. 3a). Applying both DIA- and DDA-based diGly workflows together quantified over 10,000 diGly sites in TNF-stimulated U2OS cells (Fig. 3b, Supplementary Fig. 3a, Supplementary Data 4). Both methods quantified a comparable number of ubiquitination sites (10,300 in DIA and 9500 in DDA experiment, Fig. 3b). However, the DIA experiment resulted in 248 significantly upregulated ubiquitination sites (5% FDR, median fold change 2.5), of which 37 mapped to 23 proteins known to be involved in TNF/NFκB signaling (Fig. 3c). In stark contrast, the DDA approach identified only 38 significant upregulated ubiquitination sites (5% FDR and median fold change 4.1), of which 15 mapped to 7 TNF/NFκB signaling proteins. In line with these numbers, gene ontology (GO) enrichment analysis had lower FDR values and larger group sizes for terms related to the TNF/NFκB pathway in the DIA experiment compared to DDA (Fig. 3d). Similarly, there were more significantly down-regulated ubiquitination sites (1260 in DIA vs. 517 in DDA, 5% FDR) and GOBP terms with lower FDR values in DIA than DDA experiments (Supplementary Fig. 3b and Supplementary Data 4). This large-scale downregulation of ubiquitination events may be due to the activation of deubiquitinating enzymes. In line with FDR threshold lines (Fig. 3c), power analysis exhibits lower fold-

change values (power of 0.8) for DIA compared to DDA, demonstrating increased reproducibility for DIA analysis (Supplementary Fig. 3c).

Several members of the TNF-signaling pathway have been implicated in viral infection and TNF-receptor blockage increases susceptibility to viral infection[36,37]. The 'viral processes' term was significantly enriched in our DIA analysis, in line with literature reporting the involvement of TNF-mediated ubiquitination during viral infection. Underscoring the depth of the DIA analysis, the same term failed to reach significance in the DDA analysis (Fig. 3d, Supplementary Data. 4). In agreement with previous studies, both DIA and DDA analyses revealed increased ubiquitination of prominent members of the TNF-RSC, including TRAF2, RIPK1, and BIRC2[38,39] (Fig. 3e). Increased protein ubiquitination was validated for TRAF2 and RIPK1 by western blot analysis (Supplementary Fig. 3d). DIA allowed the detection of further ubiquitination events associated with the TNF/NFκB signaling (Fig. 3c). For instance, the death domain (DD) of RIPK1 mediates interaction with FADD and TRADD[40] and we found K642 in this domain to be ubiquitinated upon TNF stimulation. Furthermore, DIA but not DDA reveals regulated ubiquitination of all members—HOIP/RNF31, HOIL-1/RBCK1, and Sharpin—of the LUBAC complex, a critical E3 ligase complex in TNF signaling[41,42] in agreement with a previous study that showed LUBAC auto-ubiquitination during inflammation[43] (Fig. 3e). p105/NFKB1, is a precursor for p50 and inhibitor of NFκB signaling[44] and we observed a striking 16-fold upregulation of K821 in its DD. Proteasome-mediated limited proteolysis of p105 during NFκB signaling yields the active p50 subunit[45–48] and the strong regulation of the K821 site suggests its involvement in this process.

DIA-based diGly analysis also uncovered TNF-regulated ubiquitination of numerous proteins known to be involved in other immune pathways. For instance, Peli2, an E3 ligase important for TLR and IL-1 signaling pathways[49] and its interaction partner TRAF6 were ubiquitinated upon TNF stimulation. We also found that STAT2, which mediates signaling by type I interferons[50], and USP13, which is involved in the antiviral response by deubiquitinating STING[51], were ubiquitinated at K161 and K3218, respectively. Our results thus suggest further molecular mechanisms for crosstalk or cross-priming function of TNF to other immune pathways during viral and bacterial infections. In summary, our DIA-based ubiquitin workflow provides an in-depth view on the dynamic ubiquitination of core and peripheral members of TNF stimulation. Apart from validating the advantages of DIA over DDA, our results provide novel regulatory ubiquitination sites, conveying a more complete picture of the various aspects of TNF signaling.

**Circadian rhythms are globally regulated by ubiquitination.** In mammals, circadian clocks are driven by interlocked transcription-translation feedback-loops. At the cellular and tissue level, they regulate oscillations of gene expression, protein abundance, and post-translational modifications[52–56]. Ubiquitination plays a pivotal role in the core clock machinery (reviewed

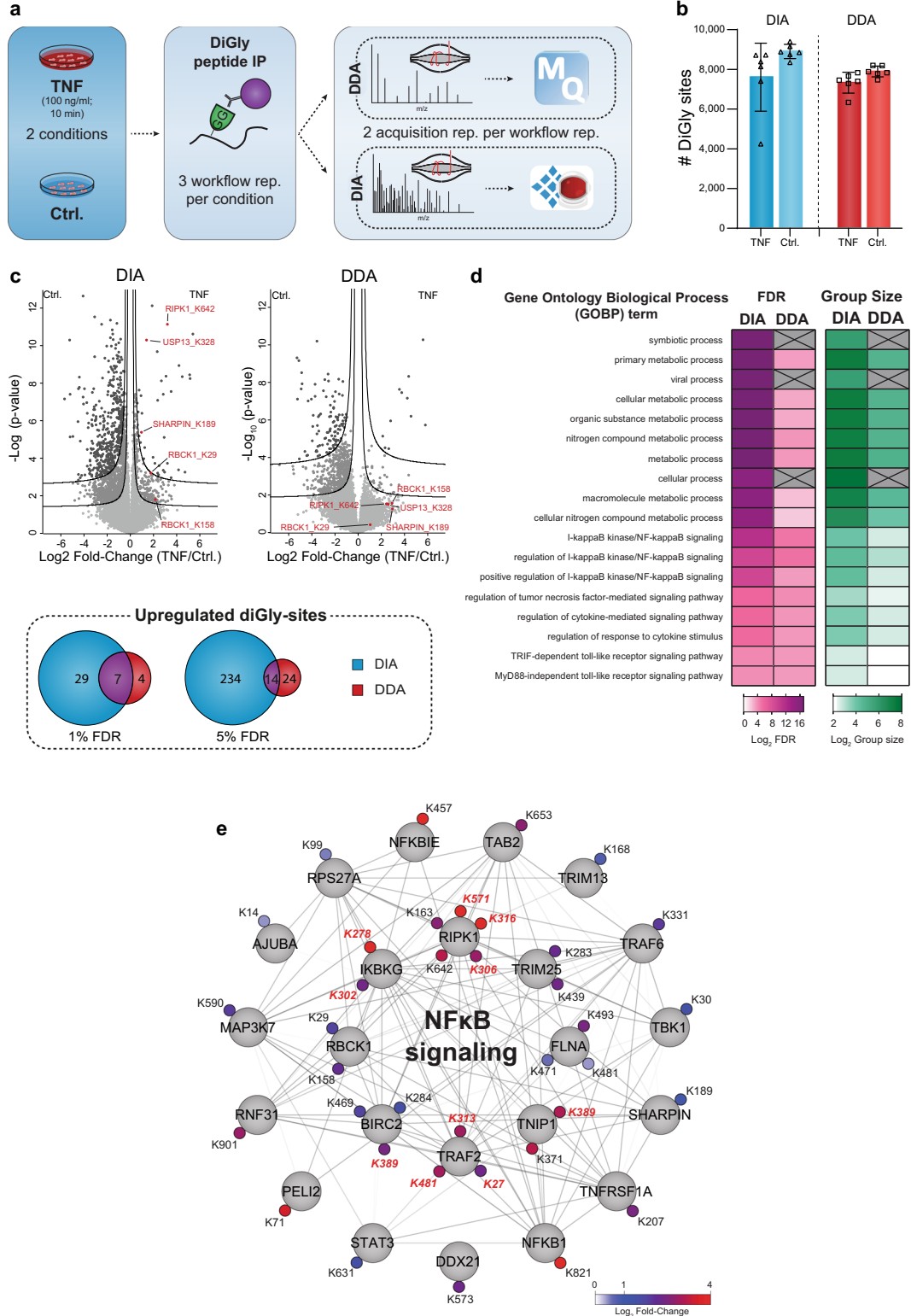

**Fig. 3 DIA enables a detailed view of the TNF-regulated ubiquitinome. a** Workflow for ubiquitinome analysis in tumor necrosis factor (TNF) signaling. **b** Identified diGly sites (±SD) for TNF treated (100 ng/ml for 10 min) and control U2OS cells in data-independent acquisition (DIA, blue) and data-dependent acquisition (DDA, red) experiments ($n = 6$, three workflow replicates measured in analytical duplicates). **c** Volcano plot of significantly regulated diGly sites at 5% false discovery rate (FDR) (FDR controlled, two-sided t-test, randomizations = 250, s0 = 0.1) (lower line) and 1% (upper line) for DIA (blue) and DDA (red) and overlaps of significantly upregulated diGly sites for 1 and 5% FDR cutoffs (t-test, s0 = 0.1). **d** Overrepresentation analysis of gene ontology biological process (GOBP) terms filtered for 5% corrected FDR (Fisher's Exact test). **e** Cytoscape network of proteins with significantly upregulated diGly sites in DIA that are associated with NFκB signaling (GO 0043122; GO 0051092; 5% FDR). Upregulated diGly sites also captured by DDA are marked in red (5% FDR). Source data are provided as a Source data file.

in ref. [57]), exemplified by the ubiquitin-dependent spatiotemporal regulation of CRY proteins, the major negative clock regulators[58]. Several studies have provided insights into ubiquitin-dependent events modulating core clock proteins and their effects[59–61]. Given the unexpected degree of phosphorylation-mediated signaling temporally regulated in vivo[52], we wondered if ubiquitination shows similar oscillations. With the high accuracy and reproducibility of our DIA-based diGly workflow, we reasoned that it would now be possible to obtain high coverage ubiquitinome quantification across a large time series sample set to answer this question.

To this end, we measured the proteome and ubiquitinome of synchronized U2OS cells—a well-established model to study the cell-autonomous circadian clock—collected every 4 h in biological quadruplicates across 32 h (Fig. 4a). Synchronization was validated by assessing the expression profile of core clock transcripts (Bmal1 and Per1) and further confirmed by PER1 and CLOCK oscillations in our proteome data (Supplementary Fig. 5a-b). After filtering for ubiquitinated peptides present in at least half the samples, we obtained 10,886 ubiquitination sites mapping to 3238 proteins (Fig. 4b, Supplementary Data 5). Measurements were highly reproducible with median Pearson coefficients >0.95 for biological replicates (Supplementary Fig. 5b-c). A total of 7590 proteins were quantified in the proteome, of which at most 143 oscillated (q-value < 0.33). This small percentage of circadian regulation at the proteome level is in line with our previous proteomics results in tissues[53] and with transcriptomics results in this cellular system[62]. Next, we normalized the intensities of the diGly peptides encompassing each ubiquitination site to their corresponding protein abundance. The resulting quantitative values represent the occupancy of the ubiquitin sites irrespective of changes in protein abundance ("Methods", Supplementary Fig. 5d).

Periodicity analysis showed that 8% of the ubiquitination sites on 18% of the proteins oscillated in a circadian manner (856 sites; 590 proteins, "Methods", q-value < 0.1, Fig. 4c, Supplementary Fig. 5e). A large proportion of rhythmic sites peaked with phases clustered around 16–20 h after synchronization (Fig. 4c and Supplementary Fig. 5e). Remarkably, 59% of these were annotated to be membrane proteins, many more than expected by chance (p < 10^{-172}; Supplementary Data 5). Overrepresentation analysis revealed that these proteins are predominantly involved in transport of small molecules, such as ions, amines, and organic acids (Fig. 4d). These findings point to a potential metabolic function of circadian membrane protein ubiquitination.

A full quarter of rhythmic ubiquitinated proteins harbored more than one oscillating site (150 sites; Fig. 4e). To investigate the spatial arrangement of them, we developed a bioinformatic proximity analysis tool (available as part of our website for browsing and analyzing the cellular ubiquitinome http://cyclingubi.biochem.mpg.de). In 17% of these proteins, rhythmic ubiquitination sites were closer together than expected by chance (p < 0.05) and 73% were annotated as membrane proteins. Interestingly, we found several examples where these adjacent sites were mostly located in regions with potential regulatory function, such as N- and C-termini, cytosolic loops, and interaction domains (Table 1). For instance, K4, K30, and K37 of the sodium independent cystine-glutamate transporter (SLC7A11, 501 aa) are rhythmically ubiquitinated with similar phases (13.8; 13.3; 13.1 h, respectively, Fig. 4f). Likewise, the potassium chloride symporter NKCC1 (SLC12A2) has a cluster of eight rhythmically ubiquitinated sites in its C-terminal domain with similar phases (K948, K958, K966, K971, K976, K983, K991, K992; Supplementary Fig. 5f). This widely expressed solute carrier plays a key role in the regulation of ionic balance and cell volume[63]. We also discovered novel oscillating ubiquitin

modifications in the MAGE domain of MAGED1, a protein that directly interacts with the core clock protein RORα, to regulate Bmal1, Rev-erbα, and E4bp4 gene expression (Fig. 4g). Interestingly, despite these rhythmic outputs neither the Maged1 transcript, protein expression nor its binding to RORα oscillate[64]. Our results now suggest that MAGED1 activity could instead be rhythmically controlled in a post-translational manner through the multiple ubiquitinations in its MAGE domain.

Together, this in-depth view of the circadian ubiquitinome, made possible by our DIA-based diGly workflow, reveals this PTM as a major regulatory mechanism driving rhythmic processes, which include essential cellular processes such as ion transport and osmotic balance.

## Discussion

We here developed a sensitive and robust DIA-based workflow, capable of identifying 35,000 diGly peptides in single-run measurements. Both the depth of coverage and the quantitative accuracy are doubled compared to otherwise identical DDA experiments. Importantly the workflow requires no extra labeling step or offline fractionation, making it streamlined and easy to implement. Furthermore, it could be used for quantification of other PTMs relying on antibody-based enrichment such as lysine acetylation and tyrosine phosphorylation. A current limitation of the DIA method is that, like for any DIA-based analysis, including phosphoproteome analysis[25,30], the best coverage and quantification is obtained with custom-made, project-specific spectral libraries. Construction of such spectral libraries requires some effort, specialized equipment for fractionation and may not always be possible for samples with low amounts such as primary cells. Alternatively, gas phase[65] or ion mobility fractionation appear to be promising strategies to simplify the workflows for project-specific spectral library construction. Furthermore, library-free approaches may also greatly simplify DIA workflows in the future. Ongoing efforts to produce prediction tools for peptide MS/MS spectra and retention times will also greatly benefit PTM analysis[66–69].

While TMT-based workflows have the advantage of multiplexing compared to DIA workflows, they require peptide fractionation after labeling for in-depth analysis, limiting throughput. In contrast, the latest advances in nanoflow liquid chromatography now increasingly allow rapid, robust, and deep DIA-based proteome and phosphoproteome profiling, which is likely applicable to DIA-based ubiquitinome analysis as well. Furthermore, the LC-MS/MS analysis of our workflow requires only a few hundred µg and it already enables the analysis of systems such as human primary cell culture models where protein material is limited. However, further sensitivity advances are limited by the initial antibody-based enrichment, which currently requires 0.5–1 mg of sample. If this step could be scaled down and the subsequent peptide purification eliminated altogether, sample amount requirements could become much smaller yet. A workflow without a peptide-clean-up step would also aid to further improve throughput and reproducibility, making the entire workflow more streamlined.

By converting from a DDA to a DIA workflow we demonstrate a dramatic increase in the number of ubiquitination sites that can consistently and significantly be quantified. Given the inherent sensitivity of our single-run approach allowing system-wide investigations of ubiquitination dynamics of biological processes, we applied it to TNF signaling. This provided an in-depth view on the ubiquitination dynamics of TNF signaling, covering core and peripheral signaling members, which a parallel DDA analysis failed to provide. Apart from validating the advantages of DIA over DDA, our results showed that like phosphorylation,

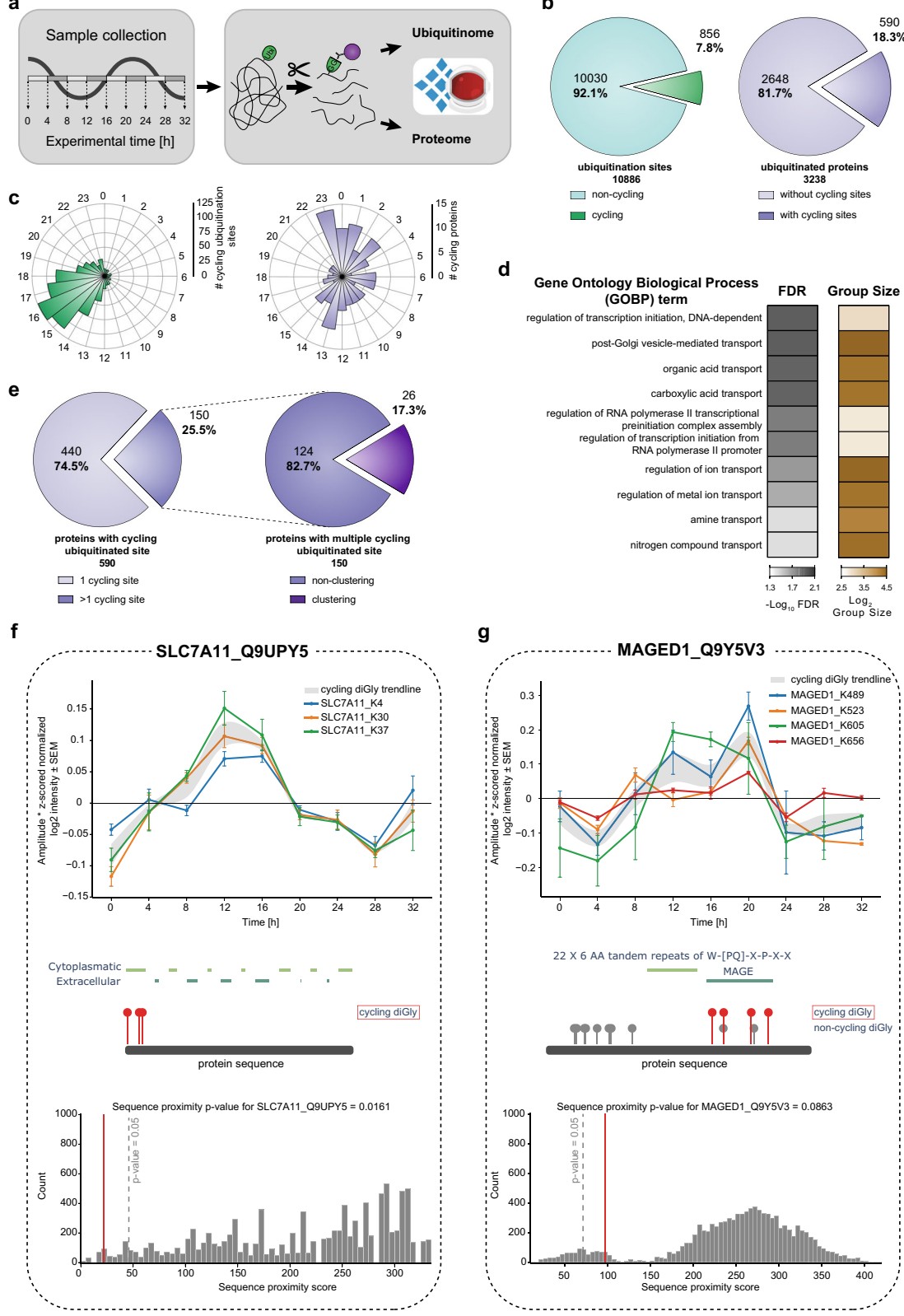

ubiquitination signaling events are rapidly induced after TNF stimulation. Unexpectedly, we still pinpointed novel TNF-regulated sites on proteins that were not previously described in this well-studied pathway. The rich resource provided here could be further explored to investigate the functions of these ubiquitination events in TNF signaling in health and disease.

System-wide circadian proteomics studies have so far been limited to the dynamic regulation of protein and phosphorylation levels—largely for technological reasons. Our in-depth quantitative diGly analysis of cell-autonomous circadian rhythms now extends those studies by providing a cell-intrinsic circadian map of ubiquitination dynamics. Quantifying more than 10,000

**Fig. 4 Quantification of the rhythmic ubiquitinome. a** Experimental workflow for rhythmic ubiquitinome analysis. **b** Proportion of oscillating ubiquitination sites ($q$-value < 0.1) quantified in >50% of all samples (left panel, green) and proteins with cycling ubiquitin sites ($q$-value < 0.1) (right panel, violet) **c** Rose plots indicate phase peaks for cycling ubiquitination sites (left panel, green) and proteins (right panel, violet). **d** Overrepresentation analysis of gene ontology biological processes (GOBP) filtered for top 10 significant terms. Significance is determined by 5% false discovery rate (FDR) (Fisher's Exact test). **e** Proportions of proteins with a single and multiple cycling ubiquitination sites (left panel) and those displaying cycling diGly site clusters (right panel). **f, g** Examples of proximity analysis of cycling ubiquitin clusters (http://cyclingubi.biochem.mpg.de). Cycling sites ($q$-value <0.1, ±SEM, $n = 4$ biologically independent experiments for each time point) (top) and their location in the protein sequence along with the domain annotation (middle) and proximity score (average distance, $p$-value < 0.1) (bottom) for **f** SLC7A11 ($p$-value = 0.0161) and **g** MAGED1 ($p$-value = 0.0863). Source data are provided as a Source data file.

**Table 1 Ubiquitination clusters with potential regulatory circadian functions.**

| Gene name | UniprotID | Proximity score (*p*-value) | Cycling ubiquitination sites | Membrane (GOCC) | Region/domain |
|---|---|---|---|---|---|
| SLC7A5 (LAT1) | Q01650 | 0.0015 | K19, K25, K30 | x | N-terminus |
| SLC16A1 (MCT1) | P53985 | 0.0036 | K216, K223, K224 | x | Cytosolic loop |
| LAYN | Q6UX15 | 0.0064 | K272, K273, K297, K311 | | |
| EPB41L5 | Q9HCM4 | 0.0073 | K508, K510 | x | |
| MYH9 | P35579 | 0.0079 | K555, K651, K760, K821 | x | Myosin motor, interaction with LIMCH1 |
| RTN4 (NOGO) | F8W914 | 0.0094 | K327, K332, K336 | x | Reticulon, C-terminus |
| ABHD17B | Q5VST6 | 0.0098 | K206, K207 | x | |
| SLC3A2 (MDU1) | F5GZS6 | 0.0153 | K114, K116 | x | |
| SLC7A11 (xCT) | Q9UPY5 | 0.0161 | K4, K30, K37 | x | N-terminus |
| PCNP | Q8WW12 | 0.0182 | K94, K96 | | |
| SCRIB (LAP4) | A0A0G2JNZ2 | 0.0184 | K53, K63 | x | N-terminus |
| PLXNB2 | O15031 | 0.0189 | K1743, K1757 | | C-terminus |
| TOM1 | O60784-2 | 0.0209 | K443, K446 | x | C-terminus |
| VLDLR | P98155 | 0.0220 | K828, K839 | x | C-terminus |
| H2AFY | O75367 | 0.0240 | K292, K295 | | Macro |
| SLC20A1 (GLVR1, PIT1) | Q8WUM9 | 0.0262 | K286, K320, K394, K399, K456 | x | Cytosolic loop |
| KSR1 | A0A0A0MQW1 | 0.0284 | K92, K101 | x | |
| HAS2 | Q92819 | 0.0311 | K73, K80 | | |
| SMARCD2 | B9EGA3 | 0.0318 | K200, K207 | | |
| TAX1BP1 (T6BP) | Q86VP1-2 | 0.0321 | K561, K571 | | |
| SLC20A2 (GLVR2, PIT2) | Q08357 | 0.0394 | K262, K272 | x | Cytosolic loop |
| HSP90AB1 (HSP90B) | P08238 | 0.0411 | K568, K577 | x | Interaction with NR3C1 |
| SLC12A2 (NKCC1) | G3XAL9 | 0.0420 | K237, K948, K958, K966, K971, K976, K983, K991, K992, K1125 | x | SLC12 |
| PCDHB5 | Q9Y5E4 | 0.0426 | K767, K784 | x | C-terminus |
| PPAP2B (LPP3) | O14495 | 0.0455 | K8, K15 | x | N-terminus |
| SCAMP1 | A0A087WXB0 | 0.0465 | K63, K71 | x | N-terminus |

Proteins with multiple cycling ubiquitination sites ($q$-value <0.1) in close proximity to each other ($p$-value < 0.05). Membrane protein annotation by Gene Ontology Cellular Compartment (GOCC) term "membrane" and region/domain classifications are derived from UniProt and manual annotation.

unique ubiquitination sites in synchronized U2OS cells, a standard cellular model in chronobiology, revealed that 8% of them—located on 18% of the quantified ubiquitinated proteins—oscillated in abundance. Many of the cycling sites match into the DIA library of untreated, rather than the library of proteasome inhibited cells suggesting they could have regulatory, non-degradative functions.

Our data reveal wide-spread rhythms of ubiquitination in membrane proteins, transporters, and receptors, all regulating major cellular processes such as cell volume, ion balance, and osmotic homeostasis. Intriguingly, often these cycling ubiquitination sites on membrane proteins are not randomly distributed over the protein sequence but rather cluster in certain regions such as the N- and C-terminus. Circadian rhythms in $Mg^{2+}$ and $K^+$ cellular levels and their transport have been reported in a range of eukaryotic cell types suggesting an evolutionary conservation of this mechanism. Moreover, $K^+$ transport is a key

mechanism driving electrical excitability oscillations in the mammalian master clock and Drosophila neurons[70,71] and in turn, plasma membrane potential feeds back to the cellular clock[72,73]. Despite their fundamental cellular role, little is known about the regulatory mechanisms controlling rhythms of ion levels and size in cells[74,75]. Our system-level data suggest that ubiquitination plays a major role in the rhythmic transport of ions and other compounds in the cell by temporally modulating the activity of membrane transporters. Such a mechanism would, for instance, explain the observation that red blood cells lose their daily electrophysiological rhythm after proteasome treatment[74].

We speculate that ubiquitin-dependent temporal regulation of transporter function for various substrates (e.g., sodium/phosphate/chloride—SLC20A1/SLC20A2, monocarboxylates—SLC16A1, sodium/potassium—ATP1A1, various amino acids—SLC3A2, SLC7A5, SLC7A11, and organic anions—ABCC3) and other receptors (e.g., TGFBR2 or PLXNB2) may serve as

temporal cellular switches to sense and respond to daily changes in nutrient availability. Interestingly, in our recent phospho-proteomics study of the synaptic compartment, we observed that many of the ubiquitination-related proteins had rhythmic phosphorylation sites[76]. This suggests an interplay between post-translational modifications that together could fine-tune daily cycles of membrane-mediated processes essential for proper cellular and tissue metabolism. Given the central role of transporters in chronopharmacology[77–79], ubiquitin-dependent dynamic regulation of specific membrane transporters is an important functional aspect to consider for drug administration and patient health, both key goals of chronotherapy. The data of our rhythmic ubiquitinome analysis is accessible at http://cyclingubi.biochem.mpg.de, opening up new avenues for mechanistic investigations.

## Methods

**Cell culture, treatment, harvest, and lysis.** HEK293 (human, DMSZ, ACC 635) and U2OS (human, American Type Culture Collection [ATCC], HTB-96) cells were cultivated in DMEM (Gibco, Invitrogen) supplemented with 10% fetal bovine serum (Gibco, Invitrogen), 100 U/ml penicillin (Gibco, Invitrogen), and 100 μg/ml streptomycin (Gibco, Invitrogen) at 37 °C in a humidified incubator with a 5% $CO_2$ atmosphere. For cell harvest, cells were washed twice with ice-cold PBS (Gibco, Invitrogen), centrifuged, snap-frozen in liquid nitrogen, and stored at −80 °C until lysis. Frozen cell pellets were lysed by adding lysis buffer (1% SDC in 100 mM Tris/HCl, pH 8.5) directly onto frozen cell pellets, followed by repeated aspiration and boiling for 5 min at 95 °C.

For proteasome inhibition, HEK293 or U2OS cells were treated with 10 μM MG132 (InvivoGen) at ~80% confluence for 4 h and successively harvested. For circadian cycle experiments, cells were synchronized, when they reached at least 90% confluence, with dexamethasone (1 μM) for 1 h. Following this, U2OS were washed once with PBS and the medium was replaced. The first time point was collected after 24 h of synchronization continuing the collection every 4 h across 32 h for each of the 4 biological replicates. Collected cells were stored and lysed as described. For TNF stimulation of U2OS cells, confluent cultures were either stimulated with 100 ng/ml TNF for 10 min or left unstimulated. Cells were washed 3× with ice-cold PBS, directly lysed with lysis buffer, and boiled for 5 min at 95 °C.

**Western blot analysis.** To validate TNF-signaling induction U2OS cells were plated in 6-well plates and when confluent stimulated for 5, 10, 15, 30, and 60 min with 100 ng/ml TNF or left untreated. After stimulation, cells were washed in PBS and lysed in 4% SDS in 100 mM Tris/HCl, pH 8. Lysates were boiled, sonicated, and protein concentrations were estimated using BCA. SDS sample loading buffer (450 mM Tris-HCl, pH 8, 60% (v/v) glycerol, 12% (w/v) SDS, 0.02% (w/v) bromophenol blue, 600 mM DTT) was added to lysates before separation on 12% Novex Tris-glycine gels (Thermo Fisher Scientific, XP00120BOX). Separated proteins were transferred onto PVDF membranes (Merck Millipore, IPVH00010). Membranes were blocked in 5% BSA in PBST and antibodies diluted in 2% BSA in PBST.

For validating increased RIPK1 and TRAF2 ubiquitination upon TNF treatment U2OS cells were either left untreated or stimulated with TNF (100 ng/ml) for 10 min, washed in PBS, and lysed in 1 ml DISC buffer (150 mM NaCl, 30 mM Tris pH 7.5, 10% glycerol, 1% Triton X-100) with protease inhibitors (Roche) and 10 mM n-ethyl-maleimide (NEM). Samples were clarified by centrifugation at 16,000 × g for 15 min, equalized to 1 mg of protein, and added directly to 20 μl packed glutathione sepharose beads pre-bound with 100 μg GST-UBA (Ubiquillin-UBA x1)[13]. Beads were incubated on a rotating wheel at 4 °C overnight, washed five times with DISC buffer, and eluted with 2× SDS sample buffer. Proteins were separated on a 10% Novex Tris-glycine gel (Thermo Fisher Scientific, XP00105BOX) and transferred onto a nitrocellulose membrane (GE Healthcare Lifescience, 10600002).

Antibodies (diluted 1:1000) used for immunoblotting were as follows: anti-phospho p65 (CST, 3033 P), anti-p65 (CST, 4764 P), anti-IκBα (CST, 9242), anti-phospho p38 (CST, 9215), anti-p38 (CST, 9212), anti-β-actin (CST, 4970) for TNF-signaling validation and anti-RIP (BD Bioscience, 610458), anti-TRAF2 (CST, 4712), and anti-β-actin (Santa Cruz, sc-47778) for validation of increased RIPK1 and TRAF2 ubiquitination.

**RNA isolation and QPCR.** RNA was isolated from three biological replicates of each U2OS time point according to manufacture instruction using the RNeasy Plus Mini Kit (QIAGEN, #74134). Isolated RNA was reversely transcribed by using first-strand cDNA synthesis kit (Thermo Fisher Scientific, #K1612). QPCR was performed at the C1000 Thermal Cycler (Bio-Rad) with iQ$^{TM}$ SYBR Green Supermix (Bio-Rad, #170-8862) with primers for *Bmal1* (froward: caggaaaaa-taggccgaatg; reverse: gcgatgacccctcttatcctg), *Per1* (forward: ggacactcctgcgaccag; reverse: gggagtgaggtggaagatctaa), and *Gapdh* (forward: agccacatcgctcagacac;

reverse: gcccaatacgaccaaatcc). The in-build analysis tool of the CFX Manager Software (Version 3.1, Bio-Rad) was used to determine the normalized expression with the ΔΔCq method of *Bmal1* and *Per1* compared to *Gapdh* in technical triplicates for all three biological replicates of each time point. The technical triplicates were further averaged and adjusted so that the highest value was set to 1. Following this, the average of all biological replicates and the SEM (standard error of the mean) was calculated for all the time points.

**Protein digestion and peptide cleanup.** Lysates were sonicated for 1 min (Branson Sonifier) and protein concentrations were estimated by tryptophan assay. After addition of CAA and TCEP to a final concentration of 10 and 40 mM, respectively, samples were incubated for 5 min at 45 °C for protein reduction and alkylation. Thereafter, Samples were digested overnight at 37 °C using trypsin (1:100 w/w, Sigma-Aldrich) and LysC (1/100 w/w, Wako).

For proteome analysis, sample aliquots (~15 μg) were desalted in SDB-RPS StageTips (Empore). Briefly, samples were first diluted with 1% TFA in isopropanol to a final volume of 200 μl. Thereafter, samples were loaded onto StageTips and sequentially washed with 200 μl of 1% TFA in isopropanol and 200 μl 0.2% TFA/2% ACN. Peptides were eluted with 60 μl of 1.25% ammonium hydroxide (NH₄OH)/80% ACN and dried using a SpeedVac centrifuge (Eppendorf, Concentrator plus). Dried peptides were resuspended in buffer A* (2% ACN/0.1% TFA) supplemented with iRT peptides (1/30 v/v) (iRT Standard, Biognosys).

For diGly peptide enrichment, samples were four-fold diluted with 1% TFA in isopropanol and loaded onto SDB-RPS cartridges (Strata™-X-C, 30 mg/3 ml or Strata™-X-C, 200 mg/6 ml, Phenomenex Inc.). Before peptide loading, cartridges were equilibrated with 8 bed volumes (BV) of 30% MeOH/1% TFA and washed with 8 BV of 0.2% TFA. Samples were loaded by gravity flow and sequentially washed twice with 8 BV 1% TFA in isopropanol and once with 8 BV 0.2% TFA/2% ACN. Peptides were eluted twice with 4 BV 1.25% NH₄OH/80% ACN and diluted with ddH₂O to a final ACN concentration of 35% ACN. Thereafter, samples were snap-frozen in liquid nitrogen, lyophilized, and stored at 4 °C until diGly peptide enrichment.

**DiGly peptide enrichment.** Lyophilized peptides were resuspended in immuno-affinity purification buffer (IAP) (50 mM MOPS, pH 7.2, 10 mM Na₂HPO₄, 50 mM NaCl) and sonicated for 2.5 min (Bioruptor plus, Diagenode). Peptide concentration was estimated by tryptophan assay. DiGly remnant containing peptides were enriched using the PTMScan® Ubiquitin Remnant Motif (K-ε-GG) Kit (Cell Signaling Technology (CST)), which was kindly provided by CST. First, antibodies were cross-linking to beads. Following Udeshi et al.[22] 1 vial of antibody coupled beads were first washed 3 times with 1 ml cold cross-linking wash buffer (100 mM sodium tetraborate decahydrate, pH 9.0), followed by 30 min incubation in 1 ml cross-linking buffer (20 mM dimethylpimipimidate cross-linking wash buffer) for 30 min at room temperature and gentle agitation. The cross-linking reaction was quenched by two consecutive washes with 1 ml cold quenching buffer (200 mM ethanolamine, pH 8.0) and 2 h incubation in 1 ml quenching buffer at room temperature under gentle agitation. After quenching cross-linked beads were washed three times with 1 ml of cold IAP and used directly for diGly peptide enrichment or stored in 1 ml 0.02% sodiumazide in phosphate-buffered saline, pH 7.4. Unless otherwise stated, 1/8 of a vial of cross-linked antibody beads and 1 mg of peptide material were used for diGly peptide enrichments. For this, peptides were added to cross-linked antibody beads and the volume was adjusted to 1 ml with IAP buffer. After 1 h of incubation at 4 °C and gentle agitation, beads were washed twice with cold IAP and five times with cold ddH₂O. For this, beads were transferred into GF-filter StageTips and for each wash step, the according wash solution was added and passed through by centrifugal force. Thereafter, GF-StageTips were stacked onto SDB-RPS StageTips and peptides were directly eluted into SDB-RPS StageTips. For this, 50 μl 0.15% TFA were added twice onto the beads and passed through by centrifugation for 5 min at 100 × g. Thereafter, 100 μl of 0.2% TFA was added on top of peptide eluates, followed by sample loading onto the stationary material of SDB-RPS StageTips. Peptides were washed, eluted, and dried as described for proteomes samples, with the difference, that 0.2% TFA was used for the first wash step. Dried peptides were resuspended in 9 μl buffer A*, supplemented with iRT peptides (1/30 v/v) for LC/MS-MS analysis.

**Basic reversed-phase fractionation.** Basic reversed-phase (bRP) fractionation for diGly peptide and proteome spectral libraries were performed on an UFLC System (Shimadzu) and EASY-nLC 1000 (Thermo Fisher Scientific, Germany), respectively.

For diGly peptide separation, lyophilized samples were resuspended in Buffer A (5 mM NH₄HCO₂/2% ACN) and 5 mg peptide material (5 mg/ml) was loaded onto a reversed-phase column (ZORBAX 300Extend-C18, Agilent). Peptides were separated at a flow rate of 2 ml/min and a constant column temperature of 40 °C using a binary buffer system, consisting of buffer A and buffer B (5 mM NH₄HCO₂/90% ACN). An elution gradient at 0% B stepwise increased to 28 in 53 min and to 78 in 6 min was deployed. Eluting peptides were automatically collected into a 96-deepwell plate while well positions were switched in 40 s intervals.

For peptide fractionation on the EASY-nLC 1000 system, ~55 μg peptide material were loaded onto a 30 cm in-house packed, reversed-phase columns (250-

μm inner diameter, ReproSil-Pur C18-AQ 1.9 μm resin [Dr. Maisch GmbH]). Peptides were separated at a flow rate of 2 μl/min using a binary buffer system of buffer A (PreOmics) and buffer B (PreOmics). An elution gradient at 3% B stepwise increased to 30% in 45 min, 60% in 17 min, and 95% in 5 min was used. Eluting peptides were concatenated into 24 fractions by switching the rotor valve of an automated concatenation system (Spider fractionator, PreOmics)[80] in 90 s intervals.

**Library sample preparation.** For individual deep diGly libraries, 2 × 5 mg of peptide was fractionated by bRP fractionation. For K48-peptide containing fraction identification, 100 μl aliquots of fractions 46 to 54 were dried in a SpeedVac, resuspended in A*, and measured on an LTQ Orbitrap XL mass spectrometer. K48-peptide containing fractions of both plates were pooled in sample pool "K48" (Supplementary Fig. 1a). Remaining fractions of both pates were concatenated into P1–P8 (Supplementary Fig. 1a), snap-frozen, and lyophilized. Lyophilized peptides were resuspended in 1 ml IAP buffer and diGly peptides were enriched as described above. In case of HEK293 library generation, an optional second supernatant IP was conducted. For this, 500 μl of previous diGly peptide enrichment supernatants were pooled as indicated (Supplementary Fig. 1a) and used for sequential diGly peptide enrichment.

For the proteome library, aliquots of U2OS samples for proteome cycling analysis were used. Approximately 3 μg peptide material of individual time points of two biological replicates, after SDB-RPS cleanup, were pooled and fractionate via bRP fractionation as described above. Fractionated samples were dried using a SpeedVac and resuspended in A* supplemented with iRT peptides (1/30 v/v) for LC-MS/MS measurement and spectral library generation.

**Nanoflow LC-MS/MS proteome measurements.** Peptides were loaded onto a 50 cm, in-house packed, reversed-phase columns (75 μm inner diameter, ReproSil-Pur C18-AQ 1.9 μm resin [Dr. Maisch GmbH]). The column temperature was controlled at 60 °C using a homemade column oven and binary buffer system, consisting of buffer A (0.1% formic acid (FA)) and buffer B (0.1% FA in 80% ACN), was utilized for low pH peptide separation. An EASY-nLC 1200 system (Thermo Fisher Scientific), directly coupled online with the mass spectrometer (Q Exactive HF-X, Thermo Fisher Scientific) via a nano-electrospray source, was employed for nanoflow liquid chromatography, at a flow rate of 300 nl/min. For individual measurements, 500 ng of peptide material was loaded and eluted with a gradient starting at 5% buffer B and stepwise increased to 30% in 95 min, 60% in 5 min, and 95% in 5 min.

The same general setup was used, for K48-peptide containing fraction identification, while the column and mass spectrometer were changed to a 20 cm column and an LTQ Orbitrap XL, respectively.

For DDA experiments, the Thermo Xcalibur (4.0.27.19) and LTQ Tune plus (2.5.5 SP2) software were used for Q Exactive HF-X and LTQ Orbitrap XL instruments, respectively. The Q Exactive HF-X was operated in Top12 mode with a full scan range of 300–1650 $m/z$ at a resolution of 60,000. The automatic gain control (AGC) was set to 3e6 at a maximum injection time of 20 s. Precursor ion selection width was kept at 1.4 $m/z$ and fragmentation was achieved by higher-energy collisional dissociation (HCD) (NCE 27%). Fragment ion scans were recorded at a resolution of 15,000, an AGC of 1e5 and a maximum fill time of 60 ms. Dynamic exclusion was enabled and set to 20 s. The LTQ Orbitrap XL was operated in Top10 mode with a full scan range of 300–1700 $m/z$ at a resolution of 60,000. Precursor ion selection width was kept at 2.0 $m/z$ and fragmentation was achieved by collision-induced dissociation (CID) (NCE 35%).

For DIA analysis, the MaxQuant Live software suite was utilized for data acquisition[81]. The full scan range was set to 300–1650 $m/z$ at a resolution of 120,000. The AGC was set to 3e6 at a maximum injection time of 60 ms. HCD (NCD 27%) was used for precursor fragmentation and fragment ions were analyzed in 33 DIA windows at a resolution of 30,000, while the AGC was kept at 3e6.

**Nanoflow LC-MS/MS diGly measurements.** DiGly peptide enriched samples were measured on a Q Exactive HF-X using the same instrumental setup as for proteome analysis. For diGly single-run measurements one quarter (2 μl) and for diGly library preparation one-half (4 μl) of enriched samples were loaded for LC-MS/MS analysis, unless stated otherwise. Loaded peptides were eluted using a gradient starting at 3% buffer B and stepwise increased to 7% in 6 min, 20% in 49 min, 36% in 39 min, 45% in 10 min, and 95% in 4 min.

For DDA analysis, the MS was operated in Top12 mode with a full scan range of 300–1350 $m/z$ at a resolution of 60,000. AGC was set to 3e6 at a maximum injection time of 20 s. Precursor ion selection width was kept at 1.4 $m/z$ and fragmentation was achieved by HCD (NCE 28%). Fragment ion scans were recorded at a resolution of 30,000, an AGC of 1e5 and a maximum fill time of 110 ms. Dynamic exclusion was enabled and set to 30 s.

For DIA analysis, the MaxQuant Live software suite was employed for data acquisition[81]. The full scan range was set to 300–1350 $m/z$ at a resolution of 120,000. The AGC was set to 3e6 at a maximum injection time of 60 ms. HCD (NCD 28%) was used for precursor fragmentation and resulting fragment ions were analyzed in 46 DIA windows at a resolution of 30,000 (unless otherwise

stated) and an AGC of 3e6. DIA window distribution parameters PdfMu and PdfSigma were set to 6.161865 and 0.348444, respectively, unless stated otherwise (Supplementary Data 6).

**Raw data analysis.** DDA raw data used for K48-peptide fraction identification and DIA and DDA comparisons were analyzed with MaxQuant (1.6.2.10) using default settings and enabled match between runs (MBR) functionality. Carbamidomethyl (C) was defined as fixed modification and Oxidation (M), Acetyl (Protein N-term), and DiGly (K) were set as variable modifications.

DDA raw data, used for spectral library construction, were processed with Spectronauts build in search engine pulsar (13.12.200217.43655)[28]. Default settings were used for proteome spectral libraries. For diGly spectral libraries, the "Best N Fragments per peptides" maximum value was adjusted to 25. For hybrid library construction DIA raw files were processed together with DDA library raw files using the same search settings.

DIA raw files were processed using Spectronaut (13.12.200217.43655)[28]. Proteome analysis was performed with default settings. For diGly analysis, diGly (K) was defined as an additional variable modification and PTM localization was enabled and set to 0. For dilution experiments, "XIC RT extraction window" was set to "static" with a window width of 10 min. Direct DIA searches used the same settings as described above.

**Bioinformatics analysis.** Data analysis was primarily performed in the Perseus software suite (1.6.7.0). For diGly site analysis, Spectronaut normal report output tables were aggregated to diGly sites using the peptide collapse plug-in tool for Perseus[30]. DiGly sites were aggregated using the linear model-based approach and filtered for a localization probability >0.5. Data sets of both acquisition strategies, DIA and DDA, were filtered to contain >50% valid values in at least one experimental condition. Missing values were imputed based on a Gaussian normal distribution with a width of 0.3 and a downshift of 1.8. Student $t$-test statistics (FDR cutoff 1% or 5%; $s0 = 0.1$) for TNF-stimulation experiments were performed in Perseus. Fisher's Exact GOBP Term enrichment of upregulated diGly sites and cycling diGly sites was performed on the pantherdb website (http://pantherdb.org/) and in perseus, respectively, with Benjamini Hochberg FDR correction enabled and set to a 5% cutoff. Network representation of upregulated diGly sites was performed with the STRING app (1.5.1) in Cytoscape (3.7.2). The power analysis was performed in R (3.6.2), using the 'pwr'[82] and 'effectsize'[83] packages. The Cohen's distance was calculated based on a fixed power of 80%, a sample size of 6 per condition and a desired significance threshold of 1%. The test was set to a "two. sample" and "two.sided" $t$-test. A fold-change threshold was subsequently estimated by multiplying Cohens's distance with the pooled standard deviation separately for each peptide. We only considered sites without missing values for this analysis.

For the cycling analysis of diGly sites, data were first filtered for diGly sites identified in at least 50% across all measurements. Proteins and diGly sites raw intensities were $log_2$ transformed and normalized by median subtraction. For diGly site protein normalization the median values of biological quadruplicates were subtracted from normalized diGly sites. Missing values of protein data for subtraction were imputed based on a Gaussian normal distribution with a width of 0.3 and a downshift of 1.8. Cycling analysis of normalized protein and diGly site data was performed as previously described, but in this case with a period time of 24.8 h[52,53]. A $q$-value cutoff of <0.1 and <0.33 was used to define cycling DiGly sites and proteins, respectively.

**Website tool.** For profile plots individual $z$-scores for each protein abundance normalized diGly site and the median $z$-score and standard error of means (SEM) were subsequently determined for each time point. The resulting median $z$-scores and SEM values were multiplied with the cycling amplitude of each diGly site (Perseus periodicity analysis output). For sequence visualization and protein domain annotation each diGly site location was mapped to the first UniProt ID of its assigned protein group and was visualized based on its respective protein sequence stored in the fasta file that was used for MS/MS data analysis (human fasta, downloaded 2015). The protein sequences for visualization were obtained using the 'fasta' functions from pyteomics[84,85]. Information about protein domains was obtained from UniProt (https://www.uniprot.org/, accessed 25.05.2020), including the following categories: 'Topological domain', 'Motif', 'Region', 'Repeat', 'Zink finger', and 'Domain [FT]'.

To evaluate whether multiple observed cycling diGly sites are located in a specific region on the protein, we performed a proximity analysis. Three different metrics were evaluated: (1) the average distance (In amino acids) between all observed cycling diGly sites, (2) the minimum distance between any two observed cycling diGly sites, and (3) the maximum distance between any two observed cycling diGly sites. The observed distance metrics were compared to the distances expected from a random distribution of the diGly sites of a protein across all of its lysines. 10,000 random distributions were considered, and an empirical $p$-value was estimated based on the fraction of random samples with a smaller or equally small distance metric as the observed cycling diGly sites. For the main analysis, diGly sites with a $q$-value ≤ 0.1 were considered as cycling diGly sites.

Data preprocessing and visualization for the dashboard was performed using the python programming language. Following libraries were utilized for data processing: numpy (1.18.1), pandas (0.24.2), re, random, and pyteomics[84,85] (4.2). Several libraries from the HoloViz (0.11.3) family of tools were used for data visualization and creation of the dashboard, including panel and holoviews (1.13.2), but also bokeh (2.0.1), plotly (4.6.0), and matplotlib (3.0.3).

**Reporting summary**. Further information on research design is available in the Nature Research Reporting Summary linked to this article.

## Data availability

All mass spectrometry data have been deposited on the ProteomeXchange Consortium via the PRIDE database with the dataset identifier PXD019854. A file linking mass spectrometry raw data in the ProteomeXchange folder to the associated experiments in the manuscript is available (Supplementary Data 7). The proximity analysis tool for the investigation of cycling diGly sites is available on http://cyclingubi.biochem.mpg.de. Information about protein domains was obtained from UniProt (https://www.uniprot.org/, accessed 25.05.2020). Source data are provided with this paper.

## Code availability

Custom code for the proximity analysis, implemented on http://cyclingubi.biochem.mpg.de has been deposited on GitHub (https://github.com/MannLabs/CyclingProximityAnalysis).

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

## Acknowledgements

F.B. and M.S.R. were supported by the Volkswagen Foundation (93071), M.S.R. also received funding from the German Research Foundation DFG (Project 329628492-SFB1321, INST 86/1800-1 FUGG, and project 428041612). We thank all the members of the Department of Signal Transduction and Proteomics, in particular, Igor Paron and Christian Deiml for MS technical assistance, Bianca Splettstoesser for technical help with experimental work, and Johannes Müller, Alex Strasser, and Lisa Schweitzer for columns. We also thank Sharah Meszaros and Steven Dewitz for cell culture assistance. We are grateful to Arno F. Alpi, Jesper Olsen, Chuna Choudhary, André Michaelis, and Jakob Bader for constructive and insightful discussions. We specially thank Roberto Polakiewicz, Florian Gnad, Sean Landry, and Cell Signaling Technology (CST) for gifting of PTMScan® Ubiquitin Remnant Motif (K-ε-GG) Kits.

## Author contributions

F.M.H., O.K., M.T., F.B., and M.S.R. designed experiments. TNF experiments were performed by F.M.H., M.T., and C.S. Circadian experiments were conducted by F.M.H. and F.B. Computational proximity analysis was performed by I.B. Data were analyzed by F.M.H. F.M.H., O.K., M.T., M.S.R., B.A.S., and M.M. wrote, reviewed, and edited the manuscript. All authors read and commented on the manuscript.

## Funding

## Competing interests

The authors declare no competing interests.
