## [Peer Review File · Nature Communications]

REVIEWER COMMENTS

Reviewer #1 (Remarks to the Author):

Hansen et al describe a workflow for the mass spec analysis of ubiquitin modification sites using Data-independent acquisition (DIA) methods. Peptides bearing the diGly signature of ubiquitin modification are enriched by tried and tested immunoprecipitation with a diGly specific monoclonal antibody. What is required for DIA approaches is a large spectral library with as complete coverage as possible and this is achieved by treating cells with the proteasome inhibitor MG132 to boost the levels of ubiquitinated species and extensively fractionating the diGly enriched peptides from these cells and from untreated cells. Using this approach the authors generate a library of almost 90,000 diGly modified peptides using the commercially available Spectronaut software. It is a large amount of work to generate this library and this should be a very useful resource for the ubiquitin community. This spectral library is the starting point for the DIA studies. The authors then optimize the run conditions for single shot DIA analysis on the very widely used Q-Exactive platform. Data are processed using the MaxQuant live software that the Mann lab have made available to the scientific community. Using these approaches the authors identify up to 35,000 diGly containing peptides in a single run. This is a large improvement on previous methodologies and again will be useful for the scientific community. While the focus of the paper was clearly on the technology development there were a number of points that should be addressed

1. In figure 3 (TNF experiment), they focus only on upregulated sites (more ubiquitinated during TNF treatment), but the bulk of the data show reduced ubiquitination (the left hand sides of the volcano plots in figure 3c). From a purely numerical position they seem to be ignoring most of their data. In terms of TNF biology this could be very important, particularly as it is clear that DUBs are regulated. Many of these sites may be modified in a very transient nature and this may vary between proteins. Certainly the downregulated sites should be subjected to the same bioinformatic analysis as the upregulated sites and commented on in the Results section. Does this make any difference to the comparison between DIA and DDA ?

2. From the fold change versus p-value plots (3c), the cutoff lines (5% FDR and 1% FDR) seem to be implying that it is reproducibility among replicates that is causing the difference between DDA and DIA. i.e the cutoffs are higher for DDA than DIA because of greater inter sample variability. They should comment on this. A Principal Component Analysis plot for replicates for DDA and DIA would help.

3 A comparison between DDA and DIA for the log₂ ratio data (Fig 3c) would be interesting. What is the correlation? In theory the underlying ratio should be consistent (same samples) but if DDA and

DIA disagree it might be worth exploring why this is. You can actually see this in some of the sites they highlight on the charts that show quite different ratios between the two methods.

4 The identification of circadian cycling ubiquitination sites on proteins that also contain non-cycling sites (fig 4g) is very interesting. It is not clear from the data if these sites are in fact on the same molecules. One possibility is that there are two different pools of the same protein: one pool that is modified with non-cycling ubiquitin (which may be regulatory rather than inducing degradation) and another pool that is modified with cycling ubiquitin (that may induce degradation). There is also the issue here of occupancy. When looking at the data in figure 4g it may be that individual molecules are only modified at a single site and that many molecules are unmodified. These points should be addressed and commented on.

In general this is well conducted research and the technical advances presented will be very useful to the scientific community.

Reviewer #2 (Remarks to the Author):

The manuscript "Data-independent acquisition method for ubiquitinome analysis reveals regulation of circadian biology" by Hansen et al. explores the use of data independent acquisition (DIA) mass spectrometry to improve the detection and quantification of ubiquitinated proteins. The authors claim that using DIA outperforms equivalent DDA analyses of the same ubiquitinome samples, and then apply the DIA-based workflow to two biological experiments: TNF stimulation which is used to compare the improvement in biological interpretation of DIA vs DDA, and circadian rhythm, which is used to showcase a time-point style experiment with DIA. The two major claims in the paper are then the improved workflow and the novel circadian rhythm-associated biological conclusions.

Overall, this manuscript is well-written and likely of general interest as the combination of a popular data acquisition method with an equally popular post-translational modification. Admittedly, neither of these on its own is novel (that is, DIA and antibody-enriched diGly peptides) and the data processing is, at this point, routine (project specific spectral libraries); with that considered, the novel aspects of the paper lie with the two benchmarking data sets themselves and the biological interpretation of the circadian rhythm dataset.

To strengthen these two original conclusions, I would suggest the authors consider the following major and minor revisions:

> Benchmarking experiments were compared to literature in the manuscript text (“Compared to recent reports in the literature (Udeshi et al 2020), these numbers double diGly peptide identifications in a single run format”), however it would greatly strengthen the authors’ claims if they directly compared their DIA approach to that “state of the art” method published just this past January 2020 in this journal (Nat Comm) by Udeshi et al. The claim that the DIA-based method is superior is weakened by not having this comparison benchmarking the two directly and the claim greatly improved by adding this comparison to Supplemental Figure 2.

> In the interest of reproducing the optimized method described in the paper, it would be helpful to provide readers a table of the windowing scheme with the start/middle/end MZ for each cycle.

> Figure 2b is not clear. The left y-axis denotes “CV” but the square vs triangle points also denote CV for a fraction of peptides. The authors might consider splitting this figure into two, rather than using the two y-axes: one figure showing the violin plots of the CVs if that authors consider this the important message, but also showing a histogram of CVs because this type of figure (CV reproducibility) is generally visualized as a histogram.

> Figure 2c shows the linearity of peptides using line of identity, but many of these dilution series don’t appear to be linear (in particular K27, K33). Typically, linearity studies for quantitative proteomics are performed using 1:3, 1:1, and 3:1 ratios rather than the serial dilution used here; serial dilutions are more commonly used for “hybrid proteome” style figures (see Navarro et al 2016, Nat Biotechnol) or calibration curves (see Pino et al 2020, J Prot Res). Rather than the R² of these lines, the authors might consider visualizing this data in the LFQBench style (Navarro et al 2016, Nat Biotechnol) or else with the %CV of each point on a secondary y-axis (see Grant & Hoofnagle 2014, Clin Chem).

> One of the authors’ main claims is on “the power of DIA for improving data completeness and sensitivity in a single run analysis format.” This reviewer agrees that DIA should have more systematic measurements and therefore less missing data than a parallel DDA run, however the authors don’t support this claim besides in total number of detections. I would recommend the authors include a quick analysis of the missing data in the DDA vs DIA experiments, such as the missingness per run and the missingness across experiments, at the peptide and the protein level. This might be appropriate in Figure 3, specifically Fig 3b where the authors compare the number of total identified and quantified diGly sites between DDA and DIA -- I suspect the DIA approach will look much more impressive than the DDA approach in terms of data completeness, and this would greatly strengthen the authors’ claim.

> In the statistical analysis of the TNF biological application, it would be helpful if the authors described how they handled missingness (assuming that there was missingness in the data, and that reasons for missingness were different between the DIA and DDA data sets). Were missing values imputed? If not, how did the authors handle missing data in the statistical analysis? (The authors do describe how they approach missingness in the cycling analysis, which prompts this question for how Perseus handled missingness.)

> The authors report no statistical power testing, but I suspect that performing a power analysis would better inform the appropriate fold-change cutoffs (Fig 3c) especially considering the increased quantitative reproducibility of DIA vs DDA which would probably allow for better (e.g. lower) FC sensitivity. I'm not sure if power analyses are performed in Perseus in addition to the FDR cutoffs included in the figure and legend, but power analyses for pairwise comparisons can be performed in MSstats (Choi et al 2014, Bioinformatics) and would provide the appropriate FC cutoff given the number of replicates, the variation in the measurements, and a predetermined statistical power (e.g. 80%).

> In the discussion, the authors mention "Library-free approaches would greatly aid to simplify DIA workflows in the future and there are considerable efforts currently being invested into producing prediction tools for MS/MS spectra and retention time." To clarify, is this specifically in the context of analyzing PTMs by DIA? Library-free approaches using prediction tools are fairly common for proteome profiling (Searle et al 2020, Nat Comm) but I agree that it is not yet possible to predict fragmentation spectra nor retention time for PTMs beside phosphorylation. Clarifying what the authors mean in this sentence would help.

> Also in the discussion, "Compared to single run TMT-based workflows, DIA suffers from lower throughput." -- This is only true if the multiplexed TMT experiment is not prefractionated prior to MS analysis. I think it's rare that TMT multiplexes aren't off-line fractionated prior to MS analysis, and so this argument is not quite representative of common workflows.

> "Furthermore, the LC MS/MS analysis of our workflow requires only a few hundred μg and it already enables analysis of systems such as human primary cell culture models where protein material is limited" -- To clarify, the "few hundred μg " is because of the off-line fractionation and spectral library generation? Chromatogram library approaches (Pino et al 2020, Mol Cell Prot) would only require a few extra μg of material in comparison, although to my knowledge chromatogram library-based approaches haven't been used for PTM analysis.

> “Intriguingly, often these cycling ubiquitination sites on these membrane proteins are not randomly distributed over the protein sequence but rather cluster in certain regions such as the N- and C-terminus.” The authors should expand on this conclusion, as it didn’t seem strongly supported by the evidence provided and even seems intuitive to me that a large moiety like ubiquitin would be attached at free domains (e.g. termini) rather than in the transmembrane domains. Additionally, I would not be surprised if the only detected regions are regulatory just because that's what is most exposed for membrane proteins, and therefore that's where the detected diGly are located. To strengthen this claim, the authors might consider a few additional analyses. For the observations (proteins) which led to this conclusion, what was the overall protein sequence coverage? Did they detect loops/partial transmembrane regions? (That is, I wouldn't expect any K or R in transmembrane regions for digestion, anyway, which might be biasing the observation of termini vs internal peptides.) It may be helpful to show for a random selection of the proteins in Table 1 the visualization from a tool like Protter (see Omasits et al 2014, Bioinformatics).

> Website (<http://cyclingubi.biochem.mpg.de/>) doesn’t connect for me. It may just need permissions enabled?

> Data repository appears to contain all appropriate files. The authors might consider including a table or key of meta-data to annotate which file names belong to which experiments, considering the impressive number of raw files represented in this work (I didn’t see this scrolling through the file names, nor in the Supplemental tables).

Reviewer #3 (Remarks to the Author):

Overall, this manuscript from Hansen et al. set an ambitious goal and will pave the way to numerous applications in biological research. Using two different cell lines as biological models, the authors described a proteomics workflow, which is enhancing the number of identified ubiquitination (diGly peptides) sites. Then, as a proof of concept, the authors applied their method to TNF-alpha signalling and investigated ubiquitination around the circadian cycle using a standard cell model in the chronobiology field (U2OS cells).

While the first part of the manuscript showing the combination of a DIA approach combined to diGly antibody-based enrichment constitute a tour de force by identifying many new diGly peptides, the second part of the paper concerns me more.

Indeed, better validations (see all my points in detail below) showing the viability of the biological model and showing the identified patterns are reliable, seem necessary.

The most convincing would be at least to show western blot (combined with IP) data on proteins showing a differential ubiquitination level upon TNF-alpha stimulation. The same stand for proteins showing rhythmic ubiquitination around the circadian cycle (see points 6 and 14 below).

There are inconsistent statements along the manuscript regarding TNF-alpha stimulation and validation of the treatment should be carefully rechecked (see points 8 and 9 below).

Regarding the validation of U2OS cells synchronization there is a discrepancy between the results of the current manuscript and the literature. Hence, the rhythmicity of additional core clock components and output (such as Cry1, Cry2, Per2, Nr1d1, Dbp, Tef, Npas2, E4bp4) should be carefully checked in addition to rechecking Bmal1 and Per1 (see point no 11 below).

Moreover, regarding the rhythmicity of the U2OS proteome, in addition to its use to normalize the ubiquitinated proteome (Figure 4), its rhythmicity should also be compared to its corresponding transcriptome. This would also help validating the current biological experimental model. Transcriptomic data are available in papers from the Hogenesch or Sehgal labs (see point 13 below)

Many hypotheses, for instance, the control of MAGED1 by multiple ubiquitination of its MAGE domain, are formulated but none is pursued further.

So, it seems necessary that these aspects be analysed deeper in a revised version of this manuscript.

Comments and specific issues.

1. Page 2 and page 20. Summary section, introduction, discussion:

This work is the first quantifying circadian ubiquitination dynamics using a mammalian cell line (U2OS). However, it is not the first study investigating cycling ubiquitination. See the following publications.

Szabo et al. 2018. Ubiquitylation dynamics of the clock cell proteome and TIMELESS during a circadian cycle. *Cell Reports*, 23(8), 2273–2282. PMID: 29791839

Wang, et al. 2018. A proteomics landscape of circadian clock in mouse liver. *Nature Communications*,9(1), 1553. PMID: 29674717

2. Page 5, figure 1 and supplementary figure 1. The figure 1a and supplementary Figure 1 are informative and clarify the description of the workflow.

3. Page 5 end of first paragraph.

“Indicating that different PTMs can act on the same sites” What is the biological significance of this, perhaps this point could be mention/discussed in the discussion?

4. Page 5 and page 6.

A certain amount of data is generated but not analysed. This is the case for the U2OS cells data treated or untreated with MG132 that are neither present in the supplementary tables nor analyzed in the paper. Perhaps, as an additional workflow validation, it would be interesting to quantify the effect of MG132 on the U2OS proteome (and the ubiquitinated proteome) and compared the analysis with already published data using a similar biological model as for instance in the following papers.

Larance et al. 2013 "Global subcellular characterization of protein degradation using quantitative proteomics." *Mol Cell Proteomics* 12(3): 638-650. PMID 23242552.

Povlsen et al. 2012. "Systems-wide analysis of ubiquitylation dynamics reveals a key role for PAF15 ubiquitylation in DNA-damage bypass." *Nat Cell Biol* 14(10): 1089-1098. PMID 23000965.

5. Page 6 and page 9, figure 2C.

The authors did not plot the data for the K29 Ubiquitin chain linkage, which is present in the supplementary table 3. Could they please add the plot and discuss if/why the correlation is less high compared to the other linkages?

6. Page 10 and page 11, TNF-alpha stimulation.

The most convincing would be to show western blot (combined with IP) data of one or several candidates among the 37 proteins showing a differential ubiquitination level upon TNF-alpha stimulation.

7. Page 10.

“Underscoring the (...) analysis (Fig. 3d, Supplementary Fig.3)”

Why is supplementary figure 3 mentioned here? Please clarify the purpose of Supplementary Figure 3.

8. Page 10, page 13, page 22 and supplementary figure 3.

Regarding the TNF-alpha stimulation. What is exactly the concentration used to stimulate the U2OS cells?

There are three different concentrations depicted in the manuscript

Page 13: 10 ng/μl for 10 min

Page 22: 100 ng/ml for 10 min in the Cell culture section

Page 22: 100 ng/ml for 10 min in the Western Blot analysis section

Supplementary figure 3 legend: 100 ng/μl for 0 5 10 15 30 60 minutes

What's the vehicle control treatment?

9. Supplementary figure 3.

Compared to the time 0 the TNF-alpha treatment leads to an increase of total p65 and this increase seems similar to the increase of phospho p65. Hence, there seems to be no effect of TNF-alpha on phospho p65.

Why is the phospho p38 non phosphorylated at 30 and 60 min while other studies found a sustained activation up to one hour.

See the figure 2A of the following paper.

Liu, C., P. Zhao, Y. Yang, X. Xu, L. Wang and B. O. Li (2016). "Ampelopsin suppresses TNF- α -induced migration and invasion of U2OS osteosarcoma cells." *Molecular Medicine Reports* 13(6): 4729-4736. PMID: 27082056

10. Page 13.

"Given the unexpected degree of phosphorylation-mediated signaling temporally regulated in vivo". Perhaps, the authors should also mention these papers here describing temporally regulated ubiquitination.

Szabo et al. 2018. Ubiquitylation dynamics of the clock cell proteome and TIMELESS during a circadian cycle. *Cell Reports*, 23(8), 2273–2282. PMID: 29791839

Wang, et al. 2018. A proteomics landscape of circadian clock in mouse liver. *Nature Communications*,9(1), 1553. PMID: 29674717.

11. Page 13, page 22 and Supplementary Figure 5a.

The protocol used by the authors in this manuscript, 1 μ M of dexamethasone for one hour, is similar to the protocol used in the following papers to synchronize U2OS cells (synchronization with 0.1 μ M of dexamethasone).

Ref 59 of the manuscript. Hughes, M. E. et al. Harmonics of circadian gene transcription in mammals. *PLoS Genet* 5, e1000442, doi:10.1371/journal.pgen.1000442 (2009).

Altman et al 2015 "MYC Disrupts the Circadian Clock and Metabolism in Cancer Cells." *Cell Metab* 22(6): 1009-1019. PMID: 26387865.

Jang, et al. (2015). "Ribosome profiling reveals an important role for translational control in circadian gene expression." *Genome Res* 25(12): 1836-1847. PMID: 26338483.

However, when looking at the data, the authors show that Bmal1 and Per1 are peaking respectively at CT12 and CT20 when these papers show a peak phase at CT0/24 for Bmal1 and CT10 for Per1. Could you explain the discrepancy between these results?

12. Page 16 and figure 4 C, Rhythm analysis.

While I was able to find the data in supplementary table 5 used to produce the left panel of figure 4C, I was not able to find the information (peak phase) regarding the left panel displaying cycling proteins.

What about the overall amplitudes of the cycling proteome and the cycling ubiquitinome in U2OS cells?

13. Page 16, figure 4c and Supplementary Figure 5: Rhythm analysis

To further validate their experimental model and as performed in Robles et al. 2014 (ref 52 of this manuscript) in mouse liver, the authors should also compare their circadian U2OS proteome dataset with circadian transcriptomics data performed in U2OS cells. Transcriptomics data are available in papers from the Hogenesch or Sehgal labs like in the following paper.

Jang, et al. (2015). "Ribosome profiling reveals an important role for translational control in circadian gene expression." *Genome Res* 25(12): 1836-1847. PMID: 26338483.

In order to find cycling proteins and cycling ubiquitinations, it might be appropriate to use at least one other standard method, for instance JTK Cycle (Hughes et al. reference 59 of the manuscript) to make sure the conclusions of the paper remain valid.

14. Page 13 to page 15.

The most convincing would be at least to show western blot (combined with IP) data for proteins showing rhythmic ubiquitination around the circadian cycle. For instance, the global ubiquitination level of protein such as SLC7A11 or NKCC1 that display synchronous cycling ubiquitinations should be checked around the circadian cycle.

15. Page 20

“Many of the cycling sites match into the DIA library of untreated, rather than the library of proteasome inhibited cells suggesting they could have regulatory, non-degradative functions.”

Perhaps this observation should be further developed and analysed in the results section than only mentioned in the discussion. What’s the proportion of cycling ubiquitinated sites differentially regulated by MG132 treatment? (see point 4).

16. Page 20 Supplementary table 5

At the ubiquitination level, the authors did not obtain data for circadian core clock proteins. Have they been quantified and/or identified? What about other circadian ubiquitination study? Is there any correlation / similarities with data from the only other mammalian circadian ubiquitination study?

Wang, et al. 2018. A proteomics landscape of circadian clock in mouse liver. *Nature Communications*,9(1), 1553. PMID: 29674717

17. Supplementary figure 4 and supplementary table 5.

Pathway or network analyses are often biased by the number of proteins in the pathway and are often biased toward well-known pathways. Did the authors consider this inherent bias in pathway generating software?

Minor comments

Overall the figures, tables and supplementary tables should be carefully checked.

The manuscript should be proofread for some typos and inconsistencies of format.

Some example found in the main text

Please choose one format between TNF or TNF-alpha

Page 6 : put CVs into brackets (CVs)

Page 12 : post-translation modifications

Page 19 : This provided and in depth view

Page 22 : anti I κ B α (CST 92424792)

Page 23 : U2OS is not always written in the same format U-2 OS

Page 23 : Trypsin(1:100..... then lysC(1/100)

Page 25 : pates around line 21

Some example found in the supplementary figures.

Supplementary figure 1.

Collections of Fractions (capital f)

Supplementary figure 2.

I guess the last sentence in the figure legend "Validation of TNF for the indication time. " is not where it should be.

Supplementary figure 3. Please indicate on the figure (or at least the figure legend) the name of the protein and the residue(s) you are looking at i.e Phospho-NF- κ B p65 (Ser536), NF- κ B p65, Phospho-p38 MAPK (Thr180/Tyr182) p38 MAPK etc...

Supplementary figure 5.

in panel A, B, D, E, F, CT (circadian time) should be used instead of time or timepoint.

Typos in the supplementary tables.

I also found many typos in the supplementary tables that should be carefully proofread.

For instance, in supplementary tables 1 and 3.

supplementary table 1 : Typos “resoultion”; “accuired”; “destributions”, “measurments” several times (B3 B4) ; Resoultion B4) , “titratrion” (colummA) “corseping”

“titrarion” etc

supplementary table 3: typos “accuired” ; “of the” in table descirption

“intesities" sheet D ☐ many iterations of “qunatification”

Point-by-point answers to ‘Data-independent acquisition method for ubiquitinome analysis reveals regulation of circadian biology’ by Hansen et al.

We are delighted that the reviewers found our paper of considerable interest. We also thank all the reviewers for their evaluation, constructive comments and the detailed examination of our work, which improved the manuscript substantially. Please find our point-by-point answers below.

REVIEWER COMMENTS

Reviewer #1:

Hansen et al describe a workflow for the mass spec analysis of ubiquitin modification sites using Data-independent acquisition (DIA) methods. Peptides bearing the diGly signature of ubiquitin modification are enriched by tried and tested immunoprecipitation with a diGly specific monoclonal antibody. What is required for DIA approaches is a large spectral library with as complete coverage as possible and this is achieved by treating cells with the proteasome inhibitor MG132 to boost the levels of ubiquitinated species and extensively fractionating the diGly enriched peptides from these cells and from untreated cells. Using this approach the authors generate a library of almost 90,000 diGly modified peptides using the commercially available Spectronaut software. It is a large amount of work to generate this library and this should be a very useful resource for the ubiquitin community. This spectral library is the starting point for the DIA studies. The authors then optimize the run conditions for single shot DIA analysis on the very widely used Q-Exactive platform. Data are processed using the MaxQuant live software that the Mann lab have made available to the scientific community. Using these approaches the authors identify up to 35,000 diGly containing peptides in a single run. This is a large improvement on previous methodologies and again will be useful for the scientific community. While the focus of the paper was clearly on the technology development there were a number of points that should be addressed.

We thank this reviewer for their positive evaluation, especially regarding ‘the large improvement on previous methodologies’ and the detailed and accurate description of our work

1. In figure 3 (TNF experiment), they focus only on upregulated sites (more ubiquitinated during TNF treatment), but the bulk of the data show reduced ubiquitination (the left hand sides of the volcano plots in figure 3c). From a purely numerical position they seem to be ignoring most of their data. In terms of TNF biology this could be very important, particularly as it is clear that DUBs are regulated. Many of these sites may be modified in a very transient nature and this may vary between proteins. Certainly the

downregulated sites should be subjected to the same bioinformatic analysis as the upregulated sites and commented on in the Results section. Does this make any difference to the comparison between DIA and DDA?

We agree that the bulk of regulated sites shows a downregulation. We have now applied the same bioinformatics analysis for downregulated sites as for upregulated sites (Supplemental Fig. 3b; Supplemental Table 4) and included this analysis in the revised manuscript (Page 10, Lines 241-245).

2. From the fold change versus p-value plots (3c), the cutoff lines (5% FDR and 1% FDR) seem to be implying that it is reproducibility among replicates that is causing the difference between DDA and DIA. i.e the cutoffs are higher for DDA than DIA because of greater inter sample variability. They should comment on this. A Principal Component Analysis plot for replicates for DDA and DIA would help.

It is indeed the increased reproducibility of the DIA approach that results in the lower FDR lines. To make this point more clear, we performed a power analysis, as also suggested by the second reviewer. This analysis confirms the increased quantitative reproducibility of DIA vs DDA. At a fixed power, significance threshold and sample size for both DIA and DDA, DIA enabled the identification of more significantly changing peptides compared to DDA at the same fold change cutoff. We have included this analysis (Supplementary Figure 3C) and commented on it in the results (Page 10, Lines 245-247).

As suggested by the reviewer, we also performed a PCA analysis, which can be seen below. To exclude effects due to imputation (as less missing values were observed in DIA compared to DDA, see Supplementary Fig. 2g and Page 6, Line 181-183), we performed the analysis on the data filtered for no missing values. While TNF stimulated and unstimulated samples clustered separately in both cases, we observed, in comparison to DDA, a somewhat higher concordance between replicates for DIA, indicating higher reproducibility and lower variability between DIA measurements, even on the sites consistently quantified by both methods.

Figure 1 PCA of TNF treated samples

Blue -> DIA, red -> DDA; square -> treated; circle -> untreated

3 A comparison between DDA and DIA for the log₂ ratio data (Fig 3c) would be interesting. What is the correlation? In theory the underlying ratio should be consistent (same samples) but if DDA and DIA disagree it might be worth exploring why this is. You can actually see this in some of the sites they highlight on the charts that show quite different ratios between the two methods.

This is a very interesting point. To address it, we compared the fold changes (TNF stimulated vs unstimulated) obtained by DIA and DDA experiments. Again, to avoid effects introduced by data imputation we performed the fold change comparison on data filtered for no missing values. All ubiquitination sites show a good correlation (Pearson correlation coefficient of 0.78, see figure below), indicating that the reported fold changes by DDA and DIA are in a good agreement with each other. In particular, significantly upregulated sites after stimulation (5% FDR, DIA data) exhibit an excellent Pearson correlation of 0.97. We agree with the reviewer that some individual sites can show different ratios in the plot. This can be due to limited reproducibility of DDA measurement, where imputed values can have an impact on reported ratios.

Figure 2 Fold change comparison between DIA and DDA

Samples were filtered for no missing values and fold changes between TNF treated and untreated samples were compared between DIA and DDA. Density shows the distribution of all sites and black dots ubiquitination sites, which were identified as significantly upregulated by DIA (5% FDR).

4 The identification of circadian cycling ubiquitination sites on proteins that also contain non-cycling sites (fig 4g) is very interesting. It is not clear from the data if these sites are in fact on the same molecules. One possibility is that there are two different pools of the same protein: one pool that is modified with non-cycling ubiquitin (which may be regulatory rather than inducing degradation) and another pool that is modified with cycling ubiquitin (that may induce degradation). There is also the issue here of occupancy. When looking at the data in figure 4g it may be that individual molecules are only modified at a single site and that many molecules are unmodified. These points should be addressed and commented on.

These points are very interesting and important for understanding the biological significance of PTMs, including ubiquitination. Unfortunately, bottom up proteomics has a limited ability to report whether identified PTM sites are present on the same molecule or on different molecules, making the discrimination between different pools of the same protein difficult. However, our quantification of the modified site is based on singly, doubly (M2), or multiply (M3) modified peptides and this is indicated by the multiplicity. We provide this information for each peptide in our online tool (<http://cyclingubi.biochem.mpg.de>). Around 3% of diGly sites in our dataset were identified along with a second diGly site modification on the very same peptide, providing valuable information of coexisting diGly site on the same molecule.

Determination of PTM site occupancy requires the identification and quantification of both, modified and unmodified, counterpart peptides. Ubiquitin site occupancy determination is particularly challenging as

modified peptides usually have missed cleavage sites due to the modification on lysine, hence having a different sequence compared to unmodified counterparts. Therefore, diGly site occupancy calculation requires more specialized workflows than we used in this study.

In general this is well conducted research and the technical advances presented will be very useful to the scientific community.

Reviewer #2:

The manuscript “Data-independent acquisition method for ubiquitinome analysis reveals regulation of circadian biology” by Hansen et al. explores the use of data independent acquisition (DIA) mass spectrometry to improve the detection and quantification of ubiquitinated proteins. The authors claim that using DIA outperforms equivalent DDA analyses of the same ubiquitinome samples, and then apply the DIA-based workflow to two biological experiments: TNF stimulation which is used to compare the improvement in biological interpretation of DIA vs DDA, and circadian rhythm, which is used to showcase a time-point style experiment with DIA. The two major claims in the paper are then the improved workflow and the novel circadian rhythm-associated biological conclusions .

Overall, this manuscript is well-written and likely of general interest as the combination of a popular data acquisition method with an equally popular post-translational modification. Admittedly, neither of these on its own is novel (that is, DIA and antibody-enriched diGly peptides) and the data processing is, at this point, routine (project specific spectral libraries); with that considered, the novel aspects of the paper lie with the two benchmarking data sets themselves and the biological interpretation of the circadian rhythm dataset.

We thank the reviewer for the positive and detailed examination of our work. The knowledgeable and insightful points raised by the reviewer have improved and strengthened the revised manuscript.

To strengthen these two original conclusions, I would suggest the authors consider the following major and minor revisions:

1> Benchmarking experiments were compared to literature in the manuscript text (“Compared to recent reports in the literature (Udeshi et al 2020), these numbers double diGly peptide identifications in a single run format”), however it would greatly strengthen the authors’ claims if they directly compared their DIA approach to that “state of the art” method published just this past January 2020 in this journal (Nat Comm) by Udeshi et al. The claim that the DIA-based method is superior is weakened by not having this comparison benchmarking the two directly and the claim greatly improved by adding this comparison to Supplemental Figure 2.

Udeshi et al. employed a TMT multiplexing approach, which reduced MS analysis time but did not substantially improve the depth of the analysis. TMT suffers from ‘ratio compression’, a well-described issue related to TMT quantitation resulting in underestimation of ratios. To overcome this accuracy bias, Udeshi et al. employed either MS3 level fragmentation (SPS-MS3) or MS2 experiments (HCD-MS2 or FAIMS-MS2) combined with a precursor isolation purity filtering method for removing interferences in TMT experiments. We lack solid experience with both TMT labeling and this particular filtering approach and do not have suitable mass spectrometers capable of MS3 fragmentation. Therefore, if we directly compare our DIA approach to their method, we would likely report lower numbers with an impaired quantitative accuracy compared to what has been reported by Udeshi et al. We therefore believe that the fairest comparison would be between the depths reported by Udeshi et al. and in our study.

2> In the interest of reproducing the optimized method described in the paper, it would be helpful to provide readers a table of the windowing scheme with the start/middle/end MZ for each cycle.

We now provided the DIA windowing scheme used in the method in Supplementary Table 6 and included a detailed information in the methods section. We would be happy to provide more detail if required.

3> Figure 2b is not clear. The left y-axis denotes “CV” but the square vs triangle points also denote CV for a fraction of peptides. The authors might consider splitting this figure into two, rather than using the two y-axes: one figure showing the violin plots of the CVs if that authors consider this the important message, but also showing a histogram of CVs because this type of figure (CV reproducibility) is generally visualized as a histogram.

We agree that Figure 2b was confusing due to two y-axes describing CV values. We split it into two panels in the revised Figure 2 (Page 9, Line 211). In our opinion, violin plots are easier to interpret, so we prefer to visualize the CV values as violin plots, but additionally as requested by this reviewer we now also represent the same data as histograms (below).

Figure 3 CV value distribution
X-axis represents CV values. Blue -> DIA; Red -> DDA

4> Figure 2c shows the linearity of peptides using line of identity, but many of these dilution series don't appear to be linear (in particular K27, K33). Typically, linearity studies for quantitative proteomics are performed using 1:3, 1:1, and 3:1 ratios rather than the serial dilution used here; serial dilutions are more commonly used for "hybrid proteome" style figures (see Navarro et al 2016, Nat Biotechnol) or calibration curves (see Pino et al 2020, J Prot Res). Rather than the R^2 of these lines, the authors might consider visualizing this data in the LFQBench style (Navarro et al 2016, Nat Biotechnol) or else with the %CV of each point on a secondary y-axis (see Grant & Hoofnagle 2014, Clin Chem).

We thank the reviewer for raising this point, which gave us the opportunity to refine our analysis. In this experiment, our intention was to quantify ubiquitin chain-linkage type specific peptides, which are of great interest in ubiquitin studies and challenging to accurately measure by DDA (as also shown in our study). Since the ubiquitin protein is highly conserved among different species, its peptides are as well. Therefore, a "hybrid proteome" approach - that would be employed to assess the fold change quantification accuracy of ubiquitin peptides from different species - is not applicable in this case. Instead, we used a serial dilution in combination with a linear regression model to compare the performance of DDA and DIA for precision. Inspired by the LFQBench approach suggested by the reviewer, we have now extended our analysis. We now report the goodness-of-fit of both data sets to the expected dilution (dotted line). For this, we used the ratios normalized to the median of the undiluted sample. Since we calculate the R^2 value based on the expected dilution (dotted line), we can use the R^2 value as a metric to compare the DDA and the DIA data directly and assess which data better represents the expected dilution curve. We have also added the CV values for the individual data points to the revised Figure 2 (Page 7, Line 191-194; Page 9, Line 211).

5> One of the authors' main claims is on "the power of DIA for improving data completeness and sensitivity in a single run analysis format." This reviewer agrees that DIA should have more systematic measurements and therefore less missing data than a parallel DDA run, however the authors don't

support this claim besides in total number of detections. I would recommend the authors include a quick analysis of the missing data in the DDA vs DIA experiments, such as the missingness per run and the missingness across experiments, at the peptide and the protein level. This might be appropriate in Figure 3, specifically Fig 3b where the authors compare the number of total identified and quantified diGly sites between DDA and DIA -- I suspect the DIA approach will look much more impressive than the DDA approach in terms of data completeness, and this would greatly strengthen the authors' claim.

Thank you for raising this point and we are glad to incorporate this data into the manuscript. We performed this comparison with data used to investigate the power of DIA in ubiquitinome analysis rather than in the context of a specific biological application as it concerns DIA analysis in general. Please see the new analysis in Supplementary Figure 2g and the description in the revised manuscript (Page 6, Lines 181-183). We did not acquire full proteome data; therefore, a comparison on protein level has not been conducted.

6> In the statistical analysis of the TNF biological application, it would be helpful if the authors described how they handled missingness (assuming that there was missingness in the data, and that reasons for missingness were different between the DIA and DDA data sets). Were missing values imputed? If not, how did the authors handle missing data in the statistical analysis? (The authors do describe how they approach missingness in the cycling analysis, which prompts this question for how Perseus handled missingness.)

We agree with the reviewer and explained the data imputation strategy in detail in the new Methods section ("Bioinformatics analysis", Page 29, lines 727-731).

7> The authors report no statistical power testing, but I suspect that performing a power analysis would better inform the appropriate fold-change cutoffs (Fig 3c) especially considering the increased quantitative reproducibility of DIA vs DDA which would probably allow for better (e.g. lower) FC sensitivity. I'm not sure if power analyses are performed in Perseus in addition to the FDR cutoffs included in the figure and legend, but power analyses for pairwise comparisons can be performed in MSstats (Choi et al 2014, Bioinformatics) and would provide the appropriate FC cutoff given the number of replicates, the variation in the measurements, and a predetermined statistical power (e.g. 80%).

It is correct that Perseus, which was used for statistical data analysis, does not perform power analyses to estimate appropriate fold-change cutoffs, but instead applies a permutation-based FDR model to identify significantly changing sites. In contrast to other approaches, this analysis does not require the selection of separate, fixed and therefore overly cautious p-value and fold-change cutoffs. Generally, lower FDR controlled threshold lines indicate lower significance boundaries, which take the fold change as well as the p-value into consideration. DIA clearly shows lower threshold lines than DDA, demonstrating increased quantitative reproducibility.

To confirm this finding, we performed a statistical power analysis according to the reviewer's suggestion (Page 10, Lines 244-247). We selected a fixed power of 80%, a sample size of 6 per condition and a desired significance threshold of 1%. Based on these parameters we calculated the minimum fold-change that would be reported as significant, given each site's specific pooled standard deviation across samples. We only considered sites without missing values for this analysis. Supplementary Figure 3c shows the fraction of sites with detectable differences across increasing fold change cutoffs. As expected, the results of this analysis confirm that a lower fold change cutoff could be applied to DIA as compared to DDA. For instance, at a \log_2 fold change cutoff of one, 89% of sites could be detected significant in DIA while only 62% could be detected significant in DDA based on the fixed parameter set.

We decided to keep the FDR controlled model that we previously used for significance determination for the main analysis. Nevertheless, the revised manuscript includes the suggested power analysis as a panel in the Supplemental Figure 3 as an alternative validation of the increased quantitative reproducibility of DIA compared to DDA.

8> In the discussion, the authors mention "Library-free approaches would greatly aid to simply DIA workflows in the future and there are considerable efforts currently being invested into producing prediction tools for MS/MS spectra and retention time." To clarify, is this specifically in the context of analyzing PTMs by DIA? Library-free approaches using prediction tools are fairly common for proteome profiling (Searle et al 2020, Nat Comm) but I agree that it is not yet possible to predict fragmentation spectra nor retention time for PTMs beside phosphorylation. Clarifying what the authors mean in this sentence would help.

Library-free approaches using prediction tools are commonly used for proteome profiling and this will further increase in the next years. As also stated by the reviewer, the current developments such as PROSIT used in Searle et al. are mostly limited to proteome analysis and not yet available for most PTMs, especially for ubiquitinated peptides. We have changed the sentence to 'Currently considerable efforts are being invested into producing prediction tools for peptide MS/MS spectra and retention times (Searle et al 2020, Nat Comm) from which PTM analysis will also greatly benefit.' (Page 19, lines 414-416).

9> Also in the discussion, "Compared to single run TMT-based workflows, DIA suffers from lower throughput." -- This is only true if the multiplexed TMT experiment is not prefractionated prior to MS analysis. I think it's rare that TMT multiplexes aren't off-line fractionated prior to MS analysis, and so this argument is not quite representative of common workflows.

The reviewer raises an important point, which we now include in the discussion of the revised manuscript (Page 19, Line 418-419).

"While TMT-based workflows have the advantage of multiplexing compared to DIA workflows, they require peptide fractionation after labeling for in depth analysis, limiting throughput."

10> “Furthermore, the LC MS/MS analysis of our workflow requires only a few hundred µg and it already enables analysis of systems such as human primary cell culture models where protein material is limited” -- To clarify, the “few hundred ug” is because of the off-line fractionation and spectral library generation? Chromatogram library approaches (Pino et al 2020, Mol Cell Prot) would only require a few extra ug of material in comparison, although to my knowledge chromatogram library-based approaches haven’t been used for PTM analysis.

Due to the nature of the enrichment methodology, PTM analysis, including ubiquitinome analysis, usually requires high amounts of starting material to reach the depth required to cover key components of signaling pathways. In contrast, our single run DIA approach in combination with a comprehensive spectral library enables us to obtain a very high depth from small peptide amounts of only a few hundred µg per sample. To build comprehensive libraries, we performed off-line fractionation, which is followed by diGly peptide enrichments of each individual fraction. This approach can require up to mgs of protein material. We agree that these sample amount requirements are still far higher compared to what is required for full proteome measurements, but our method substantially reduces the required amount per sample compared to what has been shown in previous reports.

The mentioned gas-phase libraries strategy by Pino et al., who applied it to proteome analysis, is a very interesting approach. It would certainly be of interest to the proteomics community to extend this concept towards PTM library generation. A drawback of this easily implemented method, however, is that the depth of the chromatogram library will most likely not reach the depth of sample pre-fractionation and K48-peptide exclusion prior to diGly enrichment. (Page 19, Line 411-414)

11> “Intriguingly, often these cycling ubiquitination sites on these membrane proteins are not randomly distributed over the protein sequence but rather cluster in certain regions such as the N- and C-terminus.” The authors should expand on this conclusion, as it didn’t seem strongly supported by the evidence provided and even seems intuitive to me that a large moiety like ubiquitin would be attached at free domains (e.g. termini) rather than in the transmembrane domains. Additionally, I would not be surprised if the only detected regions are regulatory just because that’s what is most exposed for membrane proteins, and therefore that’s where the detected diGly are located. To strengthen this claim, the authors might consider a few additional analyses. For the observations (proteins) which led to this conclusion, what was the overall protein sequence coverage? Did they detect loops/partial transmembrane regions? (That is, I wouldn’t expect any K or R in transmembrane regions for digestion, anyway, which might be biasing the observation of termini vs internal peptides.) It may be helpful to show for a random selection of the proteins in Table 1 the visualization from a tool like Protter (see Omasits et al 2014, Bioinformatics).

We expect that this point of concern is closely linked to the next point addressing the availability of our website. We are sorry that the website was not accessible to the reviewer at the time. If the website would have been available to the reviewer, we believe that the reviewer would appreciate our dynamic visualization tool, which allows the user to browse through the diGly dataset of cycling diGly sites. In the manuscript, we show the simplified output of the website tool for a selection of 3 proteins (Figures 4f,g

and Supplementary Figure 5f). Information about cytoplasmic, membrane and extracellular regions of proteins are directly accessible on the website. According to the reviewer's suggestion, we also improved the information content of the sequence plots by providing the amino-acid stretches that were covered by peptides in either the full-proteome or diGly dataset. Additionally, we calculated a shared protein sequence coverage among the diGly and full-proteome dataset, thus enabling the user to more critically evaluate diGly site clustering in different sequence regions. We agree with the reviewer that it is difficult to differentiate between accessibility and regulation, which is why we believe that our manual data exploration platform on the website provides a valuable tool for more in depth analyses of specific proteins of interest.

12> Website (<http://cyclingubi.biochem.mpg.de/>) doesn't connect for me. It may just need permissions enabled?

We are very sorry that the reviewer could not access the website. We experienced server instabilities, which we now solved. We also implemented a reboot system, which automatically restarts the server every 24 hours in case of any system failure.

13> Data repository appears to contain all appropriate files. The authors might consider including a table or key of meta-data to annotate which file names belong to which experiments, considering the impressive number of raw files represented in this work (I didn't see this scrolling through the file names, nor in the Supplemental tables).

We now included a detailed list of all acquired raw files with the corresponding experiment name, Spectronaut search file, Spectronaut result file, MaxQuant result files and acquisition strategy (Supplementary Table 7).

Reviewer #3:

Overall, this manuscript from Hansen et al. set an ambitious goal and will pave the way to numerous applications in biological research. Using two different cell lines as biological models, the authors described a proteomics workflow, which is enhancing the number of identified ubiquitination (diGly peptides) sites. Then, as a proof of concept, the authors applied their method to TNF-alpha signalling and investigated ubiquitination around the circadian cycle using a standard cell model in the chronobiology field (U2OS cells).

While the first part of the manuscript showing the combination of a DIA approach combined to diGly antibody-based enrichment constitute a tour de force by identifying many new diGly peptides, the second part of the paper concerns me more.

We thank the reviewer for appreciating the extent and quality of the analytical part of the work and answer their concern about the biological part below.

Indeed, better validations (see all my points in detail below) showing the viability of the biological model and showing the identified patterns are reliable, seem necessary.

The most convincing would be at least to show western blot (combined with IP) data on proteins showing a differential ubiquitination level upon TNF-alpha stimulation. The same stand for proteins showing rhythmic ubiquitination around the circadian cycle (see points 6 and 14 below).

There are inconsistent statements along the manuscript regarding TNF-alpha stimulation and validation of the treatment should be carefully rechecked (see points 8 and 9 below).

Regarding the validation of U2OS cells synchronization there is a discrepancy between the results of the current manuscript and the literature. Hence, the rhythmicity of additional core clock components and output (such as Cry1, Cry2, Per2, Nr1d1, Dbp, Tef, Npas2, E4bp4) should be carefully checked in addition to rechecking Bmal1 and Per1 (see point no 11 below).

Moreover, regarding the rhythmicity of the U2OS proteome, in addition to its use to normalize the ubiquitinated proteome (Figure 4), its rhythmicity should also be compared to its corresponding transcriptome. This would also help validating the current biological experimental model. Transcriptomic data are available in papers from the Hogenesch or Sehgal labs (see point 13 below)

Many hypotheses, for instance, the control of MAGED1 by multiple ubiquitination of its MAGE domain, are formulated but none is pursued further.

So, it seems necessary that these aspects be analysed deeper in a revised version of this manuscript.

Comments and specific issues.

1. Page 2 and page 20. Summary section, introduction, discussion:

This work is the first quantifying circadian ubiquitination dynamics using a mammalian cell line (U2OS). However, it is not the first study investigating cycling ubiquitination. See the following publications. Szabo et al. 2018. Ubiquitylation dynamics of the clock cell proteome and TIMELESS during a circadian cycle. Cell Reports, 23(8), 2273–2282. PMID: 29791839
Wang, et al. 2018. A proteomics landscape of circadian clock in mouse liver. Nature Communications, 9(1), 1553. PMID: 29674717

Yes, we agree that two other papers mentioned by the reviewer studied ubiquitination in the context of circadian cycle. Szabo et al. performed an in vivo ubiquitin labeling assay to investigate the circadian

ubiquitinated proteome of Drosophila melanogaster. They could only identify around 300 different ubiquitinated proteins in total and did not map the ubiquitination sites, making it difficult to directly compare our dataset to the data of that study. Wang et al. assayed daily dynamics of the transcriptome and proteome of mouse liver collecting samples every 3h across 48h. However, the ubiquitination analysis was done using a more limited sample set with a time resolution of 6 hours and not covering a completed 24 cycle. The limited number of time points precluded them from performing a statistical periodicity analysis. Consequently, instead of reporting oscillations they only showed that the majority of ubiquitinated proteins have higher abundance in the two time points of the light period (ZT0-ZT6). This is in agreement with our data, since in both systems the ubiquitination pattern around the clock are comparable (taking Bmal1 mRNA as internal timing reference; in mouse liver Bmal1 mRNA peaks at ZT0 and at 12h in our U2OS experiment). Thus, in U2OS the unique phase cluster of rhythmic ubiquitination (around 16h, see figure 4C) is found shortly after the Bmal1 mRNA peak, similar to the higher number of ubiquitinated proteins in liver at ZT0 and ZT6 also hours after the peak of Bmal1 transcript. These early studies shed light on how ubiquitination levels varied around the clock; however, by using the start-of-art DIA acquisition method in conjunction with the diGly enrichment workflow we reported here the first cell autonomous circadian ubiquitination analysis on a system-wide level. Nevertheless, we see the relevance of these publications and cite them on page 13 in line 295.

2. Page 5, figure 1 and supplementary figure 1. The figure 1a and supplementary Figure 1 are informative and clarify the description of the workflow.

We thank the reviewer for the appreciation of our Figure 1.

3. Page 5 end of first paragraph.

“Indicating that different PTMs can act on the same sites” What is the biological significance of this, perhaps this point could be mention/discussed in the discussion?

We thank the reviewer for addressing this aspect. We elaborated this point further in the revised manuscript (Page 5, Line 140-144).

“According to the PhosphositePlus database, 57% of the identified diGly sites were not reported before and 7.3% of them had previously been found to be acetylated or methylated, indicating that different PTMs can act on the same sites. Thus, the growing body of diGly sites can help to identify sites of potential PTM crosstalk, an important level of functional regulation of proteins”

4. Page 5 and page 6.

A certain amount of data is generated but not analysed. This is the case for the U2OS cells data treated or untreated with MG132 that are neither present in the supplementary tables nor analyzed in the paper.

Perhaps, as an additional workflow validation, it would be interesting to quantify the effect of MG132 on the U2OS proteome (and the ubiquitinated proteome) and compared the analysis with already published data using a similar biological model as for instance in the following papers.

Larance et al. 2013 "Global subcellular characterization of protein degradation using quantitative proteomics." Mol Cell Proteomics 12(3): 638-650. PMID 23242552.

Povlsen et al. 2012. "Systems-wide analysis of ubiquitylation dynamics reveals a key role for PAF15 ubiquitylation in DNA-damage bypass." Nat Cell Biol 14(10): 1089-1098. PMID 23000965.

The U2OS cell datasets treated or untreated with MG132 were generated to construct a comprehensive spectral library from which peptides are matched into single run DIA analyses. MG132 treatment is particularly useful for library generation as it boosts the number of diGly sites by inhibiting the proteasome. As our intention was not to quantitatively compare MG132 treated and untreated cells in this publication, we did not perform any replicates of that experiment, precluding us from quantifying the effect of MG132 treatment. We agree that it would be interesting to study the effect of proteasome inhibition on the U2OS proteome and the ubiquitinated proteome, especially using our new DIA-based diGly workflow therefore we may investigate this in future studies. The data from the libraries are unfortunately too large to be provided as excel spreadsheets, we instead made them available, in a ready-to-use format for future DIA studies, along with the raw data on the PRIDE repository (see data availability in the manuscript).

5. Page 6 and page 9, figure 2C.

The authors did not plot the data for the K29 Ubiquitin chain linkage, which is present in the supplementary table 3. Could they please add the plot and discuss if/why the correlation is less high compared to the other linkages?

The K29 diGly peptide (AK^{GG}IQDK) is not retained by the column material during HPLC chromatography, consequently its detection requires different sample loading conditions. Even though we adapted the LC method to enable detection of this peptide, we could only identify/quantify the K29 diGly peptide and elution peak in the DIA, but not in the DDA experiment. MaxQuant inconsistently reported some miscleaved peptides containing the K29 diGly site with the DDA data, which is why this was not included in the previous version of the manuscript. We show the K29 peptide in one of the panels of the revised Figure 2 (Page 9, line 211). The panels of this figure are now modified according to a new analysis of the data performed as response to one comment of reviewer 2.

We now determine and plot the goodness-of-fits (R^2 values) of DDA or DIA experiment points directly to the expected dilution series as well as the CV values for individually measured samples. This new data and visualization clearly shows the power and precision of DIA vs DDA measurements.

6. Page 10 and page 11, TNF-alpha stimulation.

The most convincing would be to show western blot (combined with IP) data of one or several candidates among the 37 proteins showing a differential ubiquitination level upon TNF-alpha stimulation.

We thank the reviewer for the suggestion of this orthogonal approach to confirm our results. We used UBA-coupled Sepharose beads to pull down all ubiquitinated protein in untreated and TNF-treated cells. Western blot of captured proteins revealed increased ubiquitination of RIPK1 and TRAF2, two key members of the TNF complex, upon TNF stimulation. While RIPK1 blotting showed an increase in signal in the high molecular weight region of the gel, indicative of long ubiquitin chains or multiple ubiquitination on various sites on RIPK1, TRAF2 was primarily mono- and di-ubiquitinated upon stimulation based on the gel pattern. The results are included in Supplementary Figure 3d (see below).

Figure 4 RIPK1, TRAF2 abundance was assessed by western blotting before (Input) and after UBA Pulldowns of TNF treated and untreated U2OS cells.

7. Page 10.

“Underscoring the (...) analysis (Fig. 3d, Supplementary Fig.3)”

Why is supplementary figure 3 mentioned here? Please clarify the purpose of Supplementary Figure 3.

We thank the reviewer for spotting this error. We have corrected it and added Supplementary Table 4 instead of Supplementary Figure 3.

8. Page 10, page 13, page 22 and supplementary figure 3.

Regarding the TNF-alpha stimulation. What is exactly the concentration used to stimulate the U2OS cells?

There are three different concentrations depicted in the manuscript

Page 13: 10 ng/μl for 10 min

Page 22: 100 ng/ml for 10 min in the Cell culture section

Page 22: 100 ng/ml for 10 min in the Western Blot analysis section

Supplementary figure 3 legend: 100 ng/μl for 0 5 10 15 30 60 minutes

What's the vehicle control treatment?

We thank the reviewer for noticing this disagreement on page 13 and Supplementary Figure 3 legend. We now corrected it to the precise final TNF concentration in the media (100 ng/ml). Since the vehicle was tissue culture grade water and the TNF was diluted 1:1000 in the culture media, we considered that water would not have any effect, removing the necessity to include a control treatment condition.

9. Supplementary figure 3.

Compared to the time 0 the TNF-alpha treatment leads to an increase of total p65 and this increase seems similar to the increase of phospho p65. Hence, there seems to be no effect of TNF-alpha on phospho p65.

Why is the phospho p38 non phosphorylated at 30 and 60 min while other studies found a sustained activation up to one hour.

See the figure 2A of the following paper.

Liu, C., P. Zhao, Y. Yang, X. Xu, L. Wang and B. O. Li (2016). "Ampelopsin suppresses TNF-α-induced migration and invasion of U2OS osteosarcoma cells." *Molecular Medicine Reports* 13(6): 4729-4736. PMID: 27082056

We repeated the Western blot to test TNF stimulation in U2OS cells and obtained similar results to the experiment shown previously in the manuscript (see comparisons above). While total p65 levels are increased upon TNF stimulation, p65 phosphorylation is more strongly upregulated. This confirms, consistent with previous literature, p65 phosphorylation upon TNF stimulation. As the repeat shows

stronger upregulation of phosphorylated p65 compared to total p65, we decided to exchange the blots in the revised manuscript. The new Western blot also shows phosphorylated p38 5 to 15 minutes after stimulation and a strong downregulation at 30 minutes. The discrepancy to the blot in the publication of Liu et al could account for different supplements in the culture media, divergence of cell line passage number and/or cell handling/seeding.

10. Page 13.

“Given the unexpected degree of phosphorylation-mediated signaling temporally regulated in vivo”. Perhaps, the authors should also mention these papers here describing temporally regulated ubiquitination.

Szabo et al. 2018. Ubiquitylation dynamics of the clock cell proteome and TIMELESS during a circadian cycle. Cell Reports, 23(8), 2273–2282. PMID: 29791839

Wang, et al. 2018. A proteomics landscape of circadian clock in mouse liver. Nature Communications,9(1), 1553. PMID: 29674717.

We thank the reviewer for pointing out these very interesting publications. We see the relevance of these publications and cite them on page 13 in line 295.

11. Page 13, page 22 and Supplementary Figure 5a.

The protocol used by the authors in this manuscript, 1 μ M of dexamethasone for one hour, is similar to the protocol used in the following papers to synchronize U2OS cells (synchronization with 0.1 μ M of dexamethasone).

Ref 59 of the manuscript. Hughes, M. E. et al. Harmonics of circadian gene transcription in mammals. PLoS Genet 5, e1000442, doi:10.1371/journal.pgen.1000442 (2009).

Altman et al 2015 "MYC Disrupts the Circadian Clock and Metabolism in Cancer Cells." Cell Metab 22(6): 1009-1019. PMID: 26387865

Jang, et al. (2015). "Ribosome profiling reveals an important role for translational control in circadian gene expression." Genome Res 25(12): 1836-1847. PMID: 26338483.

However, when looking at the data, the authors show that Bmal1 and Per1 are peaking respectively at CT12 and CT20 when these papers show a peak phase at CT0/24 for Bmal1 and CT10 for Per1. Could you explain the discrepancy between these results?

As the reviewer points out this synchronization protocol, as well as this cellular system, is widely used in the circadian field by many different laboratories. Thus, many publications, using similar dexamethasone treatments, have reported cycles of core clock transcripts assessed by RT-PCR, RNAseq and/or bioluminescence using Bmal1-Luc reported cell lines. However, not all these studies described concomitant phases for the same clock components. Thus, while the three studies mentioned by the reviewer, all coauthored by John Hogenesch showed similar phases, studies from other laboratories such as Gaspar et

al. eLife 2017, Shostak et al. Nat. Commu. 2016, Lee et al. PLOS Biology 2019, showed phases concurrent with our data. Importantly, in all cases the mRNA oscillation of Bmal1, the transcriptional activator, is always in antiphase to the mRNA rhythms of the transcriptional repressors (Pers or Crys), as we also report in this manuscript.

The number of cell culture passages of any immortalized cultured cell line, as the human osteosarcoma U2OS cell, can lead to a certain heterogeneity and the same cell line cultured by different laboratories may differ from each other. This has been recently demonstrated for the HeLa cell line by the group of Rudi Aebersold using a multi-omics approach (Liu et al. Nat Biotechnol. 2019). Taking this into account and since the molecular clock is strongly modulated by the metabolic state (recently reviewed by Sinturel et al. JMB 2020), diverse culture conditions and passage number of U2OS lines in different laboratories could account for divergences on the basic metabolic state which can ultimately lead to a different phase response to the dexamethasone treatment. In any case, the anti-phase rhythms of Bmal1 and Per1 transcripts after dexamethasone treatment (shown in the Supplementary Fig. 5a,b) demonstrate proper synchronization of the U2OS cells used in our study.

12. Page16 and figure 4 C, Rhythm analysis.

While I was able to find the data in supplementary table 5 used to produce the left panel of figure 4C, I was not able to find the information (peak phase) regarding the left panel displaying cycling proteins.

What about the overall amplitudes of the cycling proteome and the cycling ubiquitinome in U2OS cells?

We updated Supplementary Table 5, which now contains amplitudes and peak phases of rhythmic diGly sites and cycling proteins.

13. Page16, figure 4c and Supplementary Figure 5: Rhythm analysis

To further validate their experimental model and as performed in Robles et al. 2014 (ref 52 of this manuscript) in mouse liver, the authors should also compare their circadian U2OS proteome dataset with circadian transcriptomics data performed in U2OS cells. Transcriptomics data are available in papers from the Hogenesch or Sehgal labs like in the following paper.

Jang, et al. (2015). "Ribosome profiling reveals an important role for translational control in circadian gene expression." Genome Res 25(12): 1836-1847. PMID: 26338483.

Following this suggestion, we have compared our proteome data to the RNAseq study indicated by the reviewer. We have found that out of the 143 proteins we detect cycling in our study only 8 are reported rhythmic by RNAseq and this overlap is even smaller (4 out of the 143) if we use the cycling ribosome profiling data. This is not surprising since a number of omics circadian studies have shown low degree of intersection between the diverse cycling datasets. For example, the above mentioned paper reported that in U2OS only one third of the cycling transcripts display rhythms of translation also found by ribosome profiling. Similar results have been reported in mouse liver: the 150 high-confident transcripts showing oscillations of ribosome profiling only account for 8% of the cycling proteins (Janic et al. Genome Research

2015) and, moreover, 20% to 50% of the total rhythmic proteome oscillate independently of their corresponding transcripts (Robles et al. PLOS Genetics 2014, Mauvoisin et al. PNAS 2014). This further strengthens our findings suggesting that the extensive ubiquitination rhythms at the cellular level impose a time of day control of protein function and abundance.

In order to find cycling proteins and cycling ubiquitinations, it might be appropriate to use at least one other standard method, for instance JTK Cycle (Hughes et al. reference 59 of the manuscript) to make sure the conclusions of the paper remain valid.

Such a comparison was done in our original publication describing the cycling analysis package in Perseus (Robles et al. 2014). The results of the comparison were displayed in the panel C of Figure S1 of this paper (included again below for this reviewer). Both the JTK-cycle and the Perseus algorithm produced comparative results, in particular in the parameters important for rhythmicity estimation, phase and q-value. The value of these parameters calculated by both algorithms showed very good correlation (Pearson $r > 0.9$, see graphs below). Since this publication, the Perseus cycling algorithm has been used to assess rhythms of diverse omics in large scale datasets in various publications (Robles et al. Cell Metabolism 2017, Noya et al. Science 2019, Bruening et al. Science 2019, Adler et al. Front. Aging Neurosci. 2020) indicating the suitability of this method for assessing daily oscillations.

14. Page 13 to page 15.

The most convincing would be at least to show western blot (combined with IP) data for proteins showing rhythmic ubiquitination around the circadian cycle. For instance, the global ubiquitination level of protein such as SLC7A11 or NKCC1 that display synchronous cycling ubiquitinations should be checked around the circadian cycle.

While we understand that the reviewer would like to see an orthogonal approach to confirm diGly regulations we report, we would have to point out that immunoprecipitations of proteins of interest followed by Western blotting for the same proteins or total ubiquitin, has several drawbacks. Most importantly such experiments require reliable antibodies. Occupancies are expected to be low, which leads to a strong enrichment of unmodified proteins. This makes the detection of ubiquitinated proteins difficult and staining the Western blot for total ubiquitin after immunoprecipitation will also detect ubiquitin on proteins interacting with the protein of interest or unspecific binders to the beads.

Therefore, we used an alternative approach to demonstrate the ubiquitination of several components of the TNF signaling pathway in response to the TNF stimulation as shown above in the response to point 6 of this reviewer. Here we pulled down all ubiquitinated proteins and stained for the protein of interest. This experiment demonstrated increased levels ubiquitinated proteins of interest in response to the stimulus. However, Western blots lack quantification accuracy and oscillation patterns with amplitudes less than 2-fold (e.g. around 1.4 for SLC 7A11) in our circadian experiment would be very difficult to be detected. Furthermore, proteins also could carry non-cycling ubiquitination sites and each site can have ubiquitin chains of different length/ topology attached, which would inevitably affect the western blot quantification.

15. Page 20

“Many of the cycling sites match into the DIA library of untreated, rather than the library of proteasome inhibited cells suggesting they could have regulatory, non-degradative functions.”

Perhaps this observation should be further developed and analysed in the results section than only mentioned in the discussion. What’s the proportion of cycling ubiquitinated sites differentially regulated by MG132 treatment? (see point 4).

As explained above in the response to the point 4 of this reviewer, the MG132 treatment was performed to generate the spectral library, potentially containing ubiquitination sites mediating proteasome-dependent protein degradation. We treated the cells with MG132 to increase the number of ubiquitination sites in the library, ubiquitination events that are either not present or present at very low levels in non-treated cells. Since this treatment was not done in replicates, the experiment was not designed for this purpose, we cannot statistically estimate with high confidence the regulation of ubiquitination in response to MG132 treatment, this would require at least three biological replicates. As also explained in the manuscript, there is reason to believe that many cycling sites have regulatory, non-degradative functions. This is consistent with the observation that four times more sites were exclusively identified by matching into the untreated library compared to the treated library.

16. Page 20 Supplementary table 5

At the ubiquitination level, the authors did not obtain data for circadian core clock proteins. Have they been quantified and/or identified? What about other circadian ubiquitination study? Is there any

correlation / similarities with data from the only other mammalian circadian ubiquitination study? Wang, et al. 2018. A proteomics landscape of circadian clock in mouse liver. Nature Communications,9(1), 1553. PMID: 29674717

As transcriptional activators and repressor, core clock proteins are present in very low stoichiometry at the cellular and tissue level, absolute quantification reported extremely low copy numbers (around 1,000 copies per cell for some of them at some time point in the circadian cycle) (Narumi et al., PNAS 2016). Thus, proteomics approaches have only successfully detected them after extensive peptide or subcellular fractionation or using targeted methods (Mauvoisin et al., PNAS 2013; Robles et al., PLOS Genetics 2014; Narumi et al., PNAS 2016; Wang et al., Cell Metab. 2017). Yet we were able to quantify some clock proteins and show their oscillations in our single shot DIA proteome data. However, we did not detect any ubiquitination on clock proteins, suggesting that clock protein ubiquitination mediates proteasome-dependent degradation as reported for many of those proteins (reviewed by Stojkovic et al. Front Mol Neurosci 2014).

As also mentioned in response to point 1 from this reviewer, Wang et al. did not perform any cycling analysis on their ubiquitin data due to a limited time resolution sampling. Due to this, we consider that any comparison would not be very useful, particularly because the sample types used in both studies are very different (mouse liver vs. U2OS human cell culture).

Nevertheless, we tried to make this comparison to answer this reviewer's point. First of all, we were surprised to see that their data contained over 100 ubiquitinated peptides with no quantitative values and multiple peptides without any protein/gene annotation in their Supplementary Data 12. Applying our filtering criteria onto their dataset (using Supplementary Data 12) resulted in only 491 ubiquitinated peptides, 115 of which were also identified in our data set. This limited overlap could be also due to sequence divergence between mouse and human proteins. When we restrained our comparison to sites on proteins that are highly conserved in both species, we found proteins such as VCP (Transitional endoplasmic reticulum ATPase) with 7 ubiquitination sites reported by Wang et al., 6 of which were also found in our dataset that in addition contained another ubiquitination sites for this protein (Supplementary Table 5). In conclusion and despite the difficulty in comparing our study to published data, we consider that our results are in strong agreement with those previous studies (see also our comments to point 1 of this reviewer) and represent a novel comprehensive analysis of ubiquitination rhythms in a cell autonomous model.

17. Supplementary figure 4 and supplementary table 5.

Pathway or network analyses are often biased by the number of proteins in the pathway and are often biased toward well-known pathways. Did the authors consider this inherent bias in pathway generating software?

The reviewer addresses a very important aspect and we are aware of the inherent bias pointed out by the reviewer. For the overrepresentation analysis, we used the panther classification system (Supplementary Figure 4) and Perseus annotations (Supplementary Table 5), which are well known and frequently used tools for such enrichment analysis. To report enriched terms, we employed these tools combining a Fisher's

Exact test with Benjamini Hochberg FDR control cutoffs. To compare between DIA and DDA strategies, we used the whole human proteome as reference set for the overrepresentation analysis. For the circadian experiment, overrepresentation analysis of cycling proteins was performed against the total identified proteins of the dataset. We also show not only the FDR values but also the group size of identified ubiquitinated proteins belonging to the corresponding enrichment term in the Figure 3d, 4d. We hope that the reviewer agrees that solving this inherent bias is beyond the scope of this project, and that we performed the analysis according to current accepted and widely used procedures.

Minor comments

Overall the figures, tables and supplementary tables should be carefully checked.

The manuscript should be proofread for some typos and inconsistencies of format.

Some example found in the main text

We appreciate that the reviewer has spotted all these mistakes. We have addressed or corrected all the points listed below and carefully checked the manuscript for typos and inconsistencies in formatting.

Please choose one format between TNF or TNF-alpha

Page 6 : put CVs into brackets (CVs)

Page 12 : post-translation modifications

Page 19 : This provided and in depth view

Page 22 : anti IκBα (CST 92424792)

Page 23 : U2OS is not always written in the same format U-2 OS

Page 23 : Trypsin(1:100..... then lysC(1/100)

Page 25 : pates around line 21

Some example found in the supplementary figures.

Supplementary figure 1.

Collections of Fractions (capital f)

Supplementary figure 2.

I guess the last sentence in the figure legend “Validation of TNF for the indication time. “ is not where it should be.

Supplementary figure 3. Please indicate on the figure (or at least the figure legend) the name of the protein and the residue(s) you are looking at i.e Phospho-NF-κB p65 (Ser536), NF-κB p65, Phospho-p38 MAPK (Thr180/Tyr182) p38 MAPK etc...

Supplementary figure 5.

in panel A, B, D, E, F, CT (circadian time) should be used instead of time or timepoint.

Typos in the supplementary tables.

I also found many typos in the supplementary tables that should be carefully proofread.

For instance, in supplementary tables 1 and 3.

supplementary table 1 : Typos “resoultion”; “accuired”; “destributions”, “measurments” several times (B3 B4) ; Resoultion B4) , “titratrion” (colummA) “corrsepinding”
“titrarion” etc

supplementary table 3: typos “accuired” ; “of the” in table descirption
“intesities” sheet D ☐ many iterations of “qunatification”

REVIEWERS' COMMENTS

Reviewer #1 (Remarks to the Author):

The authors have done a very good job responding to the comments that I made. I am happy that the paper is now suitable for publication.

Reviewer #2 (Remarks to the Author):

Hansen et al. describe an approach to ubiquitinome using DIA-MS in this paper, "Data-independent acquisition method for analysis reveals regulation of circadian biology", which constitutes a considerable effort in data acquisition and analysis.

I feel the authors have addressed my original criticisms adequately, and I have only minor clarifications below that I do not feel need to be addressed with manuscript revisions. I look forward to seeing this manuscript published as another demonstration of how DIA-MS proteomics can be applied to further improve biological investigations, especially of PTMs.

> We thank the reviewer for raising this point, which gave us the opportunity to refine our analysis. In this experiment, our intention was to quantify ubiquitin chain-linkage type specific peptides, which are of great interest in ubiquitin studies and challenging to accurately measure by DDA (as also shown in our study). Since the ubiquitin protein is highly conserved among different species, its peptides are as well. Therefore, a "hybrid proteome" approach - that would be employed to assess the fold change quantification accuracy of ubiquitin peptides from different species - is not applicable in this case. Instead, we used a serial dilution in combination with a linear regression model to compare the performance of DDA and DIA for precision. Inspired by the LFQBench approach suggested by the reviewer, we have now extended our analysis. We now report the goodness-of-fit of both data sets to the expected dilution (dotted line). For this, we used the ratios normalized to the median of the undiluted sample. Since we calculate the R^2 value based on the expected dilution (dotted line), we can use the R^2 value as a metric to compare the DDA and the DIA data directly and assess which data better represents the expected dilution curve. We have also added the CV values for the individual data points to the revised Figure 2 (Page 7, Line 191-194; Page 9, Line 211).

I do appreciate the extended analysis, and especially the additional effort to improve the quantitative comparison of these methods using the serial dilution data. The R^2 value is a good comparison, but for future reference I caution against using R^2 to assess serial dilutions as the most analytically crucial part of the dilution -- the lower limits of detection and quantification -- will not be appropriately represented by an R^2 because R^2 can be highly driven by the highest concentration points. That said, I do appreciate this improved figure, thank you for implementing this suggestion.

> We expect that this point of concern is closely linked to the next point addressing the availability of our website. We are sorry that the website was not accessible to the reviewer at the time. If the website would have been available to the reviewer, we believe that the reviewer would appreciate our dynamic visualization tool, which allows the user to browse through the diGly dataset of cycling diGly sites. In the manuscript, we show the simplified output of the website tool for a selection of 3 proteins (Figures 4f,g and Supplementary Figure 5f). Information about cytoplasmic, membrane and extracellular regions of proteins are directly accessible on the website. According to the reviewer's suggestion, we also improved the information content of the sequence plots by providing the amino-acid stretches that were covered by peptides in either the full-proteome or diGly dataset. Additionally, we calculated a shared protein sequence coverage among the diGly and full-proteome dataset, thus enabling the user to more critically evaluate diGly site clustering in different sequence regions. We agree with the reviewer that it is difficult to differentiate between accessibility and regulation, which is why we believe that our manual data exploration platform on the website provides a valuable tool for more in depth analyses of specific proteins of interest.

The website is indeed now accessible, and I was able to check a few of the proteins in Table 1. I do agree that there are many proteins in this table that appear to be truly enriched for N or C terminal ubiquitinylation (SCRIB, SLC7A5), and the visualization makes this much more apparent. It is unfortunate that not many of these proteins appear to be detected in the full-proteome dataset for a more clear comparison. The proteins with poor coverage are harder to assess whether the ubiquitinylation is truly enriched at termini or whether those peptides were just more likely to be detected because their peptides could be completely digested, ionize, and fragment well enough for detection (SLC7A11, PCDHB5). While I am not convinced that this data shows that ubiquitinylation is enriched at protein termini, I do agree that the website is useful for manual curation of protein ubiquitinylation, which I believe is the main point.

Reviewer #3 (Remarks to the Author):

The authors have addressed most of if not all my concerns and the manuscript is improved. I feel that it is now acceptable for publication.

Comments and issue that I feel can be addressed at the editing level: The panel d (TRAF2 input) in supplementary figure 3 is empty (perhaps an image compression issue). A typo Page 24 line 545 "CST92424792".

Point-by-point answers to ‘Data-independent acquisition method for ubiquitinome analysis reveals regulation of circadian biology’ by Hansen et al.

We are delighted that the reviewers think that we adequately addressed their comments and find our work suitable for publication as an article in *Nature Communications*. They have helped us to improve the manuscript, for which we are grateful to the reviewers. Please find our point-by-point answers below.

REVIEWERS' COMMENTS

Reviewer #1 (Remarks to the Author):

The authors have done a very good job responding to the comments that I made. I am happy that the paper is now suitable for publication.

We are delighted that we addressed adequately this reviewers’ comments and they found our manuscript suitable to be published in Nature Communications.

Reviewer #2 (Remarks to the Author):

Hansen et al. describe an approach to ubiquitinome using DIA-MS in this paper, “Data-independent acquisition method for analysis reveals regulation of circadian biology”, which constitutes a considerable effort in data acquisition and analysis.

I feel the authors have addressed my original criticisms adequately, and I have only minor clarifications below that I do not feel need to be addressed with manuscript revisions. I look forward to seeing this manuscript published as another demonstration of how DIA-MS proteomics can be applied to further improve biological investigations, especially of PTMs.

We are delighted that we addressed adequately this reviewer’s original concerns and would like to thank them again for their constructive input on our manuscript.

(Our answer from first review): We thank the reviewer for raising this point, which gave us the opportunity to refine our analysis. In this experiment, our intention was to quantify ubiquitin chain-linkage type specific peptides, which are of great interest in ubiquitin studies and challenging to accurately measure by DDA

(as also shown in our study). Since the ubiquitin protein is highly conserved among different species, its peptides are as well. Therefore, a “hybrid proteome” approach - that would be employed to assess the fold change quantification accuracy of ubiquitin peptides from different species - is not applicable in this case. Instead, we used a serial dilution in combination with a linear regression model to compare the performance of DDA and DIA for precision. Inspired by the LFQBench approach suggested by the reviewer, we have now extended our analysis. We now report the goodness-of-fit of both data sets to the expected dilution (dotted line). For this, we used the ratios normalized to the median of the undiluted sample. Since we calculate the R^2 value based on the expected dilution (dotted line), we can use the R^2 value as a metric to compare the DDA and the DIA data directly and assess which data better represents the expected dilution curve. We have also added the CV values for the individual data points to the revised Figure 2 (Page 7, Line 191-194; Page 9, Line 211).

I do appreciate the extended analysis, and especially the additional effort to improve the quantitative comparison of these methods using the serial dilution data. The R^2 value is a good comparison, but for future reference I caution against using R^2 to assess serial dilutions as the most analytically crucial part of the dilution -- the lower limits of detection and quantification -- will not be appropriately represented by an R^2 because R^2 can be highly driven by the highest concentration points. That said, I do appreciate this improved figure, thank you for implementing this suggestion.

We thank the reviewer for raising this point. We also think that the extended analysis improved the figure and strengthened our point on the quantitative difference between DIA and DDA. We understand the reviewer’s concern with regard to the use of the R^2 value to assess serial dilutions and agree that one needs to be careful with its use and interpretation, because of the reasons pointed out by the reviewer.

Nevertheless, as we only want to report and compare the goodness-of-fit of both DDA and DIA data sets to the expected dilution we believe that the use of the R^2 values as metric to assess which dataset better represents the expected curve is sufficient for our purpose. We now made this special purpose of the R^2 value, the direct comparison between DIA and DDA more clear by inserting ‘as a mean to directly compare the performance of DIA against DDA,’ (p7, line 183).

(Our answer from first review): We expect that this point of concern is closely linked to the next point addressing the availability of our website. We are sorry that the website was not accessible to the reviewer at the time. If the website would have been available to the reviewer, we believe that the reviewer would appreciate our dynamic visualization tool, which allows the user to browse through the diGly dataset of cycling diGly sites. In the manuscript, we show the simplified output of the website tool for a selection of 3 proteins (Figures 4f, g and Supplementary Figure 5f). Information about cytoplasmic, membrane and extracellular regions of proteins are directly accessible on the website. According to the reviewer’s suggestion, we also improved the information content of the sequence plots by providing the amino-acid stretches that were covered by peptides in either the full-proteome or diGly dataset. Additionally, we calculated a shared protein sequence coverage among the diGly and full-proteome dataset, thus enabling the user to more critically evaluate diGly site clustering in different sequence regions. We agree with the reviewer that it is difficult to differentiate between accessibility and regulation, which is why we believe that our manual data exploration platform on the website provides a valuable tool for more in depth

analyses of specific proteins of interest.

The website is indeed now accessible, and I was able to check a few of the proteins in Table 1. I do agree that there are many proteins in this table that appear to be truly enriched for N or C terminal ubiquitylation (SCRIB, SLC7A5), and the visualization makes this much more apparent. It is unfortunate that not many of these proteins appear to be detected in the full-proteome dataset for a more clear comparison. The proteins with poor coverage are harder to assess whether the ubiquitylation is truly enriched at termini or whether those peptides were just more likely to be detected because their peptides could be completely digested, ionize, and fragment well enough for detection (SLC7A11, PCDHB5). While I am not convinced that this data shows that ubiquitylation is enriched at protein termini, I do agree that the website is useful for manual curation of protein ubiquitylation, which I believe is the main point.

We are delighted that the reviewer appreciates our online tool for the manual curation of protein ubiquitination in the context of the circadian rhythm. We agree with the reviewer that we do not show that ubiquitination in general is enriched at protein termini, but rather provide several examples that appear to be specifically ubiquitinated at protein termini and more importantly an online data exploration platform, which will allow researchers to look for any protein of interest and check if diGly site clustering happens in a specific sequence region. As the reviewer also stated, we believe these are the main points here. In line with this comment, we have revised our statements in the manuscript, particularly with regards to anything that may suggest global enrichment of cycling ubiquitination sites at protein termini by saying ‘we found several examples where’ (p14, line 363).

Reviewer #3 (Remarks to the Author):

The authors have addressed most of if not all my concerns and the manuscript is improved. I feel that it is now acceptable for publication.

Comments and issue that I feel can be addressed at the editing level: The panel d (TRAF2 input) in supplementary figure 3 is empty (perhaps an image compression issue). A typo Page 24 line 545 “CST92424792”.

We are delighted that we addressed the comments to the reviewers’ satisfaction and corrected the remaining issues. We are especially grateful for this reviewer’s detailed examination of our manuscript and that they find our manuscript to be much improved.